# Chemical Profile and Biological Activities of *Brassica rapa* and *Brassica napus* Ex Situ Collection from Portugal

**DOI:** 10.3390/foods13081164

**Published:** 2024-04-11

**Authors:** Carmo Serrano, M. Conceição Oliveira, V. R. Lopes, Andreia Soares, Adriana K. Molina, Beatriz H. Paschoalinotto, Tânia C. S. P. Pires, Octávio Serra, Ana M. Barata

**Affiliations:** 1Instituto Nacional de Investigação Agrária e Veterinária (INIAV, I.P.), Av. da República, 2780-157 Oeiras, Portugal; andreia.soares@iniav.pt; 2LEAF—Linking Landscape, Environment, Agriculture and Food—Research Center, Instituto Superior de Agronomia, Associated Laboratory TERRA, Universidade de Lisboa, Tapada da Ajuda, 1349-017 Lisboa, Portugal; 3Centro de Química Estrutural, Institute of Molecular Sciences, Instituto Superior Técnico, Universidade de Lisboa, 1049-001 Lisboa, Portugal; conceicao.oliveira@tecnico.ulisboa.pt; 4Banco Português de Germoplasma Vegetal (BPGV), Agrária e Veterinária, Quinta de S. José, S. Pedro de Merelim, 4700-859 Braga, Portugal; violeta.lopes@iniav.pt (V.R.L.); octavio.serra@iniav.pt (O.S.); anamaria.barata@iniav.pt (A.M.B.); 5Centro de Investigação de Montanha (CIMO), Instituto Politécnico de Bragança, Campus de Santa Apolónia, 5300-253 Bragança, Portugal; amolina@ipb.pt (A.K.M.); tania.pires@ipb.pt (T.C.S.P.P.); 6Laboratório Associado Para a Sustentabilidade e Tecnologia em Regiões de Montanha (SusTEC), Instituto Politécnico de Bragança, Campus de Santa Apolónia, 5300-253 Bragança, Portugal

**Keywords:** *Brassica rapa*, *Brassica napus*, glucosinolates, phenolic compounds, antibacterial activity, antioxidant activity

## Abstract

This study aimed to analyse the chemical profile and biological activities of 29 accessions of *Brassica rapa* (turnips) and 9 of *Brassica napus* (turnips and seeds) collections, maintained ex situ in Portugal. HPLC-HRMS allowed the determination of glucosinolates (GLS) and polyphenolic compounds. The antioxidant and antimicrobial activities were determined by using relevant assays. The chemical profiles showed that glucosamine, gluconasturtiin, and neoglucobrassin were the most abundant GLS in the extracts from the turnip accessions. Minor forms of GLS include gluconapoleiferin, glucobrassicanapin, glucoerucin, glucobrassin, and 4-hydroxyglucobrassin. Both species exhibited strong antioxidant activity, attributed to glucosinolates and phenolic compounds. The methanol extracts of *Brassica rapa* accessions were assessed against a panel of five Gram-negative bacteria (*Enterobacter cloacae*, *Escherichia coli*, *Pseudomonas aeruginosa*, *Salmonella enterica* subsp. enterica serovar, and *Yersinia enterocolitica*) and three Gram-positive bacteria (*Bacillus cereus*, *Listeria monocytogenes*, and *Staphylococcus aureus*). The extracts exhibited activity against *S. enterica* and *S. aureus*, and two showed inhibitory activity against *E. coli* and *Y. enterocolitica*. This study provides valuable insights into the chemical composition and biological properties of *Brassica rapa* and *Brassica napus* collections in Portugal. The selected accessions can constitute potential sources of natural antioxidants and bioactive compounds, which can be used in breeding programs and improving human health and to promote healthy food systems.

## 1. Introduction

Crops of the *Brassica* L. genus, belonging to the Cruciferae family, are of great economic interest to humans as food or condiments in many cultures. In 2014, the Brassica market was worth approximately USD 14 billion [1], and it is projected to reach USD 39.40 billion by 2023 [2].

In 2018, the Brassica production reached 69,381,155 tonnes globally, further increasing to approximately 96 million tonnes in 2020 [3]. Europe accounted for 18% of the world’s production, with 9,749,374 tonnes [4], whereas in 2021, Portugal produced 285.4 tonnes of major Brassica crops [5].

Brassicas are believed to have originated in countries around the Mediterranean basin and have since spread throughout central and southwestern Asia, including the Mediterranean, Irano-Turanian, and Saharo-Sindian phytogeographic regions [6,7]. The Brassicaceae family comprises a wide range of species, valued for different plant parts, including leaves (such as kale), roots (such as turnip and radish), seeds (such as rapeseed), buds (such as Brussels sprouts), and flowers (such as cauliflower) [8].

Brassica seeds, due to their high concentration of GLS, have been used as a spicy food condiment (such as Dijon mustard in France). Additionally, the seeds have become popular for non-food applications, particularly in the cosmetic and pharmaceutical industries. Mustard seeds are used in various folk remedies in both traditional and modern medicine [9,10]. While the roots, leaves, and green parts are for human consumption. *B. rapa* L. includes turnips, leafy greens (such as bok choy, napa cabbage, mizuna, tatsoi, rapini, *grelos*, and choy sum), and oilseed crops, such as turnip rape, and yellow sarsons as well as weedy forms (*B. rapa* ssp. *sylvestris (Lam.) Briggs*) which may be wild or feral [11,12]. The Iberian leafy crop, known as *grelos* or *nabiza*, belongs to the same subspecies as Italian rapini.

*B. napus* L. is the world’s largest source of cooking oil [13], known as rapeseed, and its by-products are used in animal feed. Some cultivars of *B. napus*, such as *B. napus* var. *napobrassica*, produce edible roots. This annual vegetable has vertical roots and alternate leaves, often glaucous and with long roots, and is commonly known as Swedish turnip.

*B. rapa* spp. *rapa* are vegetables usually consumed due to their nutritional benefits and low-calorie content. Its roots present various shapes (spherical to triangular), and colours (white to purple), and have green pubescent leaves. These vegetables have a spicy flavour and sulphurous aroma due to the degradation of GLS into isothiocyanates (ITCs), which are sulphur-containing compounds. The ITCs have various biological properties and potential health benefits [14,15]). Sulforaphane and glucobrassicin are attributed to anticancer effects [16,17,18,19] and inflammatory effects [20], while progoitrin has been associated with antithyroid effects [21]. The allyl isothiocyanate and glucosinolate compounds are responsible for antioxidant, antibacterial, and antifungal properties [22,23]. Polyphenol compounds found in the Brassicaceae family may regulate metabolism, weight, chronic diseases, and cell proliferation [24]. Flavonoid compounds such as quercetin, kaempferol, and rutin are responsible for antioxidant activity and anti-inflammatory properties [25]. Turnip varieties may contain terpenoids with antioxidant, anti-inflammatory, and antimicrobial effects. However, the specific chemical profiles and biological activities can vary depending on factors such as genotype, growing conditions, and plant part analysed.

It is an important crop in traditional farming systems and in the Portuguese diet [26]. The year-round cultivation of Brassica is due to the presence of mild winters and cool summers [27].

Generations of farmers have cultivated and selected various species, resulting in the development of regional varieties known as landraces [28]. These landraces are highly adapted to their growing locations and serve essential functions in the local diet [29]. The local names of the landraces vary by region in Portugal. Different landraces may have the same name or distinct names may refer to the same landrace [30].

Portuguese Brassica landraces have high genetic variability and robustness to high and low temperatures and low humidity. Their bioactivity and resistance to biotic stresses make them a valuable germplasm source for their ethnobotanical importance and genes for breeding programs [31,32,33,34].

Germplasm banks that respond to society’s needs include this group of crops in their priorities, and the last decade of the previous century gained notoriety due to the prominence of diets, food wheels, and pyramids in consumer decisions [35]. The Portuguese Brassica collection comprises 1271 landrace accessions, categorised into three crops: 139 *B. napus* accessions, 381 *B. rapa* accessions, and 751 *B. oleracea* accessions. These accessions are collected from farms where traditional crops are grown, and farmers cultivate local varieties (landraces).

The work presented by [36] is the most recent on local breeds in Italy. In Portugal, in the 2000s, there was research into traditional varieties of the cultivated group *B. oleracea* from the Brassica collection preserved ex situ at the BPGV. However, there has been no systematic evaluation of methanol extracts in traditional Portuguese varieties of *B. rapa* since the mid-2000s [37]. There are no studies on the varieties of *B. napus* from Portugal in this area. The GLS content of Brassica landrace accessions varies depending on various factors such as the species, cultivar, plant part, climatic conditions, agronomic practices, insect attack, and microorganism intrusion [18]. Therefore, high-resolution tandem mass spectrometry (HRMS) can provide detailed information about the mass and structure of GLS non-nutritive compounds, allowing accurate identification. Furthermore, assessing their biological activity can provide valuable information about effective antimicrobial compounds for breeding aims.

## 2. Material and Methods

### 2.1. Plant Material

A collection of 29 accessions of *B. rapa*, and 9 *B. napus* species (Table 1), local varieties (landraces), conserved in the Portuguese Germplasm Bank (BPGV) and collected in the northern and central regions of Portugal, were cultivated under the same environmental conditions in Braga, North-west of Minho province during 2019, 2020, and 2021.

The plot of land remained the same. However, sowing in trays with a 1:1 soil:turf substrate was carried out on different dates between the years and species: *B. rapa* (November 2018 and March 2020), *B. napus* (February 2021). For *B. rapa*, transplantation occurred in February 2019 and April 2020, with harvesting taking place in June of both years. For *B. napus*, transplantation was carried out in March 2021, and the seeds were harvested in July and August.

The study was conducted without fertilisation, and drip irrigation was applied once a week.

Freeze-drying of the turnip samples was performed in a freeze-dryer (Scanvac Cool Safe, Labogene Scandinavian by Design). The seeds were dried in an oven. The dried material (turnip and seeds) was ground in a mill (IKA® MF 10.2, Burladingen, Germany) using a 1.0 mm thick sieve. The turnip samples were stored under vacuum in plastic film packaging with a high barrier to gases and water vapour, composed of the combination of low-density polyethylene (LDPE 60 µm)/oriented polyamide (PA 30 µm) polymers (Amcor Flexibles Portugal), and were placed in desiccators until further analysis.

### 2.2. Methanol Extracts

To identify the GLS and phenolic compounds present in *Brassica rapa* and *Brassica napus* samples, the freeze-fried material (5 g) was shaken with methanol (Merck, Darmstadt, Germany) at a ratio of 1:20 (*w*/*v*) solid-to-liquid, in an orbital shaker (Unitronic-OR, Selecta, Barcelona) at 25 °C, stirring at 70 rpm, for 2 h. The extracts were centrifuged at 3100 rpm, 5 °C, and 20 min (Sigma and Laborzentrifugen, 1k15). The supernatant was separated from the residue, filtered through a 0.45µm porous membrane, and evaporated under vacuum (40 °C, 178 mbar) in a rotary evaporator (Buchi R-114 Rotavapor Vap System, Merck, Lisboa, Portugal). The residue of extraction was weighed and placed in amber glass flasks (25 mL), at freeze (−80 °C) to be used in all analyses. The assays were performed in duplicate.

### 2.3. Liquid Chromatography and Tandem Mass Spectrometry

Three aliquots of each extract of turnip roots were analysed on a UHPLC Elute system coupled in line with a quadrupole time-of-flight Impact II mass spectrometer equipped with an ESI source (Bruker Daltonics, Bremen, Germany). High-resolution mass spectra were acquired in the ESI negative mode. Internal calibration was achieved with a solution of ammonium formate (Fisher Chemical (Hampton, NH, USA) 10 µM introduced to the ion source via a 20 µL loop at the beginning of each analysis, using a six-port valve. The mass spectrometric parameters were set as follows: end plate offset, 500 V; capillary voltage, −2.5 kV; nebulizer, 40 psi; dry gas, 8 L·min^−1^; dry temperature, 200 °C; range *m*/*z* 100–1000; and acquisition mode: data-dependent analysis (DDA) with a rate of 5 Hz, a fixed cycle time of 3 s, with an isolation window of 0.03 Da. Data acquisition and processing were performed using the Data Analysis 5.2 software.

Chromatographic separation was achieved on a Kinetex C18 column 100 Å (150 × 2.1 mm, 2.6 μm particle size, Phenomenex) at 40 °C, using a flow rate of 0.3 mL·min^−1^. The mobile phase was 0.1% *v*/*v* acid formic in water and in acetonitrile (Fisher Chemical Hampton, NH, USA), the elution gradient was as follows: 0–2 min linear gradient to 7% B; 2–23 min linear gradient to 100% B; 23–27 min isocratic 100% B; 27–30 min linear gradient to 7% B; and then the column was re-equilibrated with 7% B for 7 min.

### 2.4. Evaluation of Biological Activities

#### 2.4.1. Antioxidant Activity

Radical-scavenging capacity assay (DPPH^•^)

The scavenging effect of the DPPH^•^ free radical was assessed spectrophotometrically by the modified method of [39]. A volume of 0.1 mL of each plant extract was added at different concentrations to 2 mL of 0.07 mmol L^−1^ DPPH^•^ (Sigma, Sternheim, Germany) in 950 g L^−1^ ethanol (Merck, Darmstadt, Germany), and the mixture was shaken and stayed for 60 min at room temperature in the dark. The absorbance of samples was measured at 517 nm, using a UV–visible spectrophotometer (double-beam; Hitachi U-2010, Cincinatti, Illinois, USA). The absorption of a blank sample containing the same amount of ethanol and DPPH^•^ solution acted as a negative control. The results for the antioxidant activity were expressed as Trolox (Sigma, Sternheim, Germany) equivalent (TE) on the basis of the standard curve y = −0.0011 x + 0.0109 (r^2^ = 0.999) in terms of micromole equivalents of Trolox per g of plant dry residue extract (µmol TE g^−1^ plant dry residue extract), and a linearity range 23.97–800 µmol TE g^−1^ and an absorbance range of 0.037–0.85 AU. All determinations were performed in triplicate.

Ferric-ion-reducing antioxidant power assay

Ferric-ion-reducing antioxidant power (FRAP) measures the formation of a blue-coloured Fe^2+^-tripyridyltriazine compound from the colourless oxidised Fe^3+^ form by the action of electron-donating antioxidants. The FRAP assay was carried out using a modified methodology of [40]. Briefly, the FRAP reagent was prepared with 1 mmolL^−1^ TPTZ (Fluka, Buchs, Germany) and 2 mmol L^−1^ ferric chloride (Fluka, Buchs, Germany) in 0.25 mol L^−1^ sodium acetate (Fluka, Buchs, Germany) (pH 3.6). Diluted extracts (in 500 g L^−1^ methanol; 200 µL) were mixed with 1.8 mL FRAP reagent, allowed to stand for 4 min at room temperature (20 °C), and the absorbance of the blue complex was monitored at 593 nm and determined against a water blank, using a UV–visible spectrophotometer (double-beam; Hitachi U-2010). A standard curve y = 0.0213x + 0.0057 (r^2^ = 0.998) was prepared using different concentrations of iron sulphate (Panreac, Barcelona, Spain) (linearity range 1.25–50 µmol L^−1;^ and an absorbance range of 0.02–1.03 AU). FRAP values are presented as µmol Fe^2+^ g^−1^ of the sample (ferric reducing power). All determinations were performed in triplicate.

Total phenolic content by Folin–Ciocalteu reagent

Total phenolic content (TPC) in methanol extracts was estimated using the Folin–Ciocalteu colorimetric method described by [41] and modified by [42], using gallic acid as the standard phenolic compound. Briefly, 1–3 mL of diluted samples (methanol extracts were diluted in water to fit the standard curve), were added to 10 mL volumetric flasks containing distilled water and Folin–Ciocalteu (Sigma, Sternheim, Germany) phenol reagent (0.5 mL) and shaken. After 5 min, 1.5 mL of a 200 g L^−1^ sodium carbonate (BDH, Poole, UK) solution was added and the volume was adjusted to 10 mL with distilled water, mixed and allowed to stand for 2 h. A blank reagent using distilled water was prepared. The absorbance was measured at 750 nm using a UV–visible spectrophotometer (double-beam; Hitachi U-2010, USA). The concentrations of total phenolic compounds in the different extracts were determined based on the standard curve y = 88.003x + 0.0288 (r^2^ = 0.998) in terms of grams per litter of Gallic acid (Sigma, Sternheim, Germany) equivalents (GAE). The linearity range for this assay was determined as 6.3 × 10^−4^−1.3 × 10^−2^ gL^−1^ GAE, and an absorbance range of 0.08–1.13 AU. All determinations were performed in triplicate.

#### 2.4.2. Antibacterial Activity

The extracts were tested against foodborne contaminants: five Gram-negative bacteria, namely *Enterobacter cloacae* (ATCC 49741), *Escherichia coli* (ATCC 25922), *Pseudomonas aeruginosa* (ATCC 9027), *Salmonella enterica* subsp. *enterica serovar* (ATCC 13076), *Yersinia enterocolitica* (ATCC 8610) and three Gram-positive bacteria, namely *Bacillus cereus* (ATCC 11778), *Listeria monocytogenes* (ATCC 19111) and *Staphylococcus aureus* (ATCC 25923). All microorganisms were obtained from Frilabo, Porto, Portugal. To assess antibacterial activity, all microorganisms were incubated at 37 °C in TSB (Tryptic Soy Broth) fresh medium for 24 h before analysis to ensure they were in the exponential growth phase.

The minimum inhibitory concentration (MIC) and minimum bactericidal concentration (MBC) were determined using the microdilution method and a colourimetric assay following the protocol described by [43]. The samples were dissolved in 5% (*v/v*) DMSO (Sigma, Sternheim, Germany) and autoclaved water to achieve a final concentration of 20 mg mL^−1^. Then the samples were serially diluted to obtain different concentrations ranging from 10 to 0.15 mg mL^−1^. Initially, 100 μL of each extract was added to the first well of a 96-well microplate, which already contained 90 μL of the medium, followed by successive dilutions. Afterwards, 10 μL of inoculum (standardised at 1.5 × 10^6^ CFU/mL) was added to all wells. 

Two negative controls were prepared, one with Tryptic Soy Broth (TSB) obtained from Biolab^®^ (Hungary) and another one with the extract. Two positive controls were prepared with TSB and each inoculum and another with medium, antibiotics, and bacteria. Ampicillin and Streptomycin (Sigma, St. Louis, MO, USA), were used at 10 mg mL^−1^ and 1 mg mL^−1^, respectively, for all bacteria tested and Methicillin (Sigma, St. Louis, MO, USA) was also used at 1 mg mL^−1^ for *S. aureus*. The microplates were then covered and incubated at 37 °C for 24 h. MIC was detected by adding 40 μL of *p*-iodonitrotetrazolium chloride (Sigma, St. Louis, MO, USA) (INT, 0.2 mg mL^−1^) followed by incubation at 37 °C for 30 min. MIC was defined as the lowest concentration inhibiting visible bacterial growth indicated by a colour change from yellow to pink. MBC was determined by plating 50 μL of suspension from each well without colour change on blood agar solid medium and incubating at 37 °C for 24 h. The lowest concentration that yielded no growth determined the MBC, defined as the lowest concentration required to kill the bacteria.

### 2.5. Statistical Analysis

The results were submitted to one-way analysis of variance (ANOVA) using multiple comparison tests (Tukey HSD) to identify differences between groups. Statistical analyses were tested at a 0.05 level of probability. The range, mean, and relative standard deviation (RSD) of each parameter were calculated using the software, StatisticaTM 12.0 [44].

Statistical analysis for GLS compounds was performed in R programming language v4.1.2 [45]. Raw data were initially explored for distribution and completeness. Replicates for each accession were averaged and then grouped per geographic origin using the R package “tidyverse” [46] and plotted using the package “ggplot2” [47]. Principal component analysis (PCA) was calculated with R function prcomp and the biplot was generated using the package “ggplot2” [47]. Accessions were clustered using package “metan” [48] based on their Euclidean distance, followed by UPGMA hierarchical clustering. The dendrogram was generated with a basic R plot function.

## 3. Results and Discussion

### 3.1. GLS’ Profile by Liquid Chromatography–Tandem Mass Spectrometry

A high-resolution tandem mass spectrometry analysis was employed to identify the GLS compounds present in the methanolic extracts of turnip roots. The compounds were identified based on the accurate *m*/*z* values of the deprotonated molecules. The probable ionic formulas were validated by extracting the ion chromatograms from the raw data. The accurate mass, isotopic pattern, and fragmentation paths were evaluated and compared to available literature data [49]. MS fragmentation of glucosinolates presented two typical pathways, one associated with the common moiety (aglycone) and the other providing useful ions for the identification of the R-side chain (Figure 1). The latter, corresponds to a combined loss of sulphur trioxide and the anhydroglucose [M-H-242]^−^, leading to fragment (F1) assigned to [R + SCNOH]^−^. F1 is common to all GLSs and allows its differentiation. Furthermore, fragments F2a and F2b can also be useful to distinguish each GLS. The loss of 196 u and 193 u from the GLS precursor ions yields F2a and F2b, the fragment ions which correspond to [R + C_2_NHO_3_S]^−^ and [R + CS_2_O_3_]^−^, respectively. The fragmentation path to produce F2a or F2b is dependent on the R group present in the GLS structure. As a result, the tandem mass spectrometric analysis of GLSs generally produces F2a or F2b, but not both. Table 2 summarises the more abundant GLSs present in the extracts. The bold marks indicate the F1, F2a or F2b fragments identified in the tandem mass spectra.

The methanolic extracts also contain small amounts of hydroxycinnamic acids and their derivatives. The more relevant compounds are also listed in Table 2. As additional data, a fragmentation pathway for the precursor ion of gluconasturtiin is illustrated in Appendix A.

### 3.2. In Vitro Antioxidant Activity and Total Phenolic Content

FRAP analysis for seed turnip extracts showed a strong antioxidant activity (ranging from 2.99 to 3.18 mmol Fe^2+^g^−1^ DW), higher than root extracts (varying from 44.01–130.83 µmol Fe^2+^g^−1^ DW). The DPPH^•^ free radical scavenging activity followed the same trend of reducing power analysis, for seeds, the values ranged from 246.44 to 330.88 TEmmol·g^−1^ and for turnips 94.00 to 232.53 TEmmol·g^−1^ (Table 3 and Table 4). The antioxidant activity was attributed to the presence of glucosinolate compounds as reported by [50]. The highest antioxidant activity for seeds, obtained by the FRAP and DPPH^•^ methods, was found for the accession harvested in Braga (BPGV02578) while the highest TPC was obtained for the accession harvested in Viana do Castelo (BPGV02740). Regarding turnips, the maximum value of FRAP and DPPH^•^ was obtained for the accession collected in Viseu (BPGV06207), whereas for TPC it was for the accession (BPGV12914) collected in Aveiro (Table 3 and Table 4).

TPC obtained for seed extracts (24.90–27.39 GAE·g^−1^ DW) of the *B. napus* revealed statistically higher values than those found for turnips extracts (5.38–18.00 GAE·g^−1^ DW). The phenol content obtained for the turnip extracts for *B. rapa* and *B. napus* species was lower than those reported by [50] for Rutabaga species. The opposite was reported by other authors [51], where the TPC content found for methanolic extracts of *B. rapa* turnips was lower than the phenol values we found for the same species. Romani and colleagues [25] obtained similar results for *Brassica rapa* L. subsp, sylvestris L.

Such differences might be due to the soil and climate conditions which can affect the content of secondary metabolites in plants, or the methodologies used to extract compounds such as the period to collect the samples, conservation of the plant (fresh or dry), and extraction methods (type of solvent and concentration ratio of the plant to solvent), such as phenolic compounds [52].

The FRAP assay showed a positive correlation with the total phenolic content (y = 0.12x + 0.81; R^2^ = 0.79). However, there was only a weak positive correlation between the DPPH• free radical scavenging activity and TPCs. This difference was explained by [53] since the TPC and FRAP methods determine only hydrophilic antioxidants, while DPPH^•^ detects those soluble in organic solvents, especially alcohols. Furthermore, small molecules may have a better chance of accessing the DPPH^•^ radical due to steric inaccessibility showing higher values than larger molecules [54].

### 3.3. Antibacterial Activity

The results of the antibacterial activity, for the accessions collected in 2019, are summarised in Table 5. Each sample has been tested against bacteria such as *E. cloacae*, *E. coli*, *P. aeruginosa*, *S. enterica*, *Y. enterocolitica*, *B. cereus*, *L. monocytogenes*, and *S. aureus*. The sample BPGV02906, demonstrated bacteriostatic activity against *E. coli* and *S. enterica* with an MIC of 5 mg mL^−1^ and showed notable activity against *S. aureus* with an MIC of 2.5 mg mL^−1^. However, it was not effective against *E. cloacae*, *P. aeruginosa*, *Y. enterocolitica*, *B. cereus*, and *L. monocytogenes*, with MIC values above 10 mg mL^−1^. The sample BPGV06987B, exhibited bacteriostatic activity against *E. coli* with an MIC of 10 mg mL^−1^ and against *S. enterica* and *S. aureus* with an MIC of 5 mg mL^−1^ and 10 mg mL^−1^, respectively, but it was ineffective against the other tested bacteria with MIC values greater than 10 mg mL^−1^. Accession BPGV06987A presented a similar result to accession BPGV06987B showing bacteriostatic activity against *E. coli*, *S. enterica*, and *S. aureus* with MICs of 10 mg mL^−1^, 5 mg mL^−1^ and 5 mg mL^−1^, respectively. Accession BPGV16127, showed bacteriostatic activity against *E. coli*, *S. enterica*, and *Y. enterocolitica* with MICs of 10 mg mL^−1^, 5 mg mL^−1^, and 10 mg mL^−1^, respectively, and notable activity against *S. aureus* with an MIC of 2.5 mg mL^−1^. Accession BPGV07480, showed similar activity patterns as BPGV16127, with MIC values above 10 mg mL^−1^ for most bacteria, except for *S. enterica* (MIC 2.5 mg mL^−1^) and *S. aureus* (MIC 2.5 mg mL^−1^), where it demonstrated better inhibitory effects. Accession BPG12296 presented MIC values for *E. cloacae*, *B. cereus*, and *L. monocytogenes* greater than 10 mg mL^−1^, while for *E. coli* and *S. aureus* the MICs were 10 mg mL^−1^ and 5 mg mL^−1^, respectively. *S. enterica* and *Y. enterocolitica* had MIC values of 5 mg mL^−1^ and 10 mg mL^−1^, suggesting some inhibitory potential. Accession BGV12376 has presented MIC values indicating no inhibition for *E. cloacae*, *B. cereus*, and *L. monocytogenes* at concentrations above 10 mg mL^−1^, while the same sample has presented moderate inhibition for *E. coli* and *Y. enterocolitica* (MIC 10 mg mL^−1^) and better activity against *S. enterica* and *S. aureus* with MICs of 5 mg mL^−1^. Accession BPGV05888 was ineffective against all bacteria tested with MICs above 10 mg mL^−1^ and presented inhibition for *S. enterica* (MIC 5 mg mL^−1^) and good activity against *S. aureus* with MICs of 2.5 mg mL^−1^. Accession BPGV03990 had an MIC above 10 mg mL^−1^ for *E. cloacae*, *Y. enterocolitica*, *P. aeruginosa*, *B. cereus*, and *L. monocytogenes*, indicating no effective inhibition, while it showed good activity against *E. coli*, *S. enterica*, and *S. aureus* with MICs of 10 mg mL^−1^, 2.5 mg mL^−1,^ and 2.5 mg mL^−1^, respectively. Accession BPGV11177 was able to inhibit the growth of *E. coli* and *S. aureus* at concentrations of 10 mg mL^−1^ and 2.5 mg mL^−1^, respectively. *S. enterica* and *Y. enterocolitica* had MICs of 5 mg mL^−1^ and 10 mg mL^−1^, suggesting some degree of susceptibility. However, the sample was not effective against *E. cloacae*, *P. aeruginosa*, *B. cereus*, and *L. monocytogenes*, as the MIC was greater than 10 mg mL^−1^ for these bacteria. The pattern for accession BPGV05875 was similar to BPGV11177, with some effectiveness against *E. coli* (MIC 10 mg mL^−1^) and *S. aureus* (MIC 2.5 mg mL^−1^). *S. enterica* and *Y. enterocolitica* also showed MICs of 5 mg mL^−1^, indicating that the sample had some bacteriostatic capacity. Yet, it remained ineffective against the other bacteria, with MIC values exceeding 10 mg mL^−1^. Accession BPGV07301 was effective in inhibiting *S. aureus* at a concentration of 2.5 mg mL^−1^. The sample also inhibited the growth of *S. enterica* at 2.5 mg mL^−1^ and showed some effect on *E. coli* and *Y. enterocolitica* with MICs of 10 mg mL^−1^. Similar to the previous samples, it did not inhibit *E. cloacae*, *P. aeruginosa*, *B. cereus*, and *L. monocytogenes*, with MIC values above 10 mg mL^−1^. Accession BPGV11111 showed varying levels of inhibition against different bacteria. The Minimum Inhibitory Concentration (MIC) for *E. cloacae*, *P. aeruginosa*, *B. cereus*, and *L. monocytogenes* was determined to be greater than 10 mg mL^−1^, which suggests that the sample was not effective in inhibiting these particular bacteria. On the other hand, *E. coli* had an MIC of 10 mg mL^−1^, indicating a moderate inhibitory effect. *S. enterica* was in some whey more susceptible, with an MIC of 5 mg mL^−1^. Notably, *S. aureus* was quite responsive to the sample, with an MIC of 2.5 mg mL^−1^, indicating a good level of inhibition. The results for accession BPGV12405 were similar to BPGV11111 in that there was no inhibitory effect against *E. cloacae*, *P. aeruginosa*, *B. cereus*, and *L. monocytogenes*, as indicated by the MIC values which were again greater than 10 mg mL^−1^. *E. coli* and *S. enterica* showed MICs of 10 mg mL^−1^ and 5 mg mL^−1^, respectively, and *S. aureus* showed a promising MIC of 2.5 mg mL^−1^. It failed to inhibit the growth of *E. coli* effectively, with an MIC greater than 10 mg mL^−1^. However, the sample was effective against *S. enterica* with an MIC of 5 mg mL^−1^ and was quite effective against *S. aureus* with an MIC of 2.5 mg mL^−1^. Accession BPGV12239 demonstrated bacteriostatic activity against *E. coli* at an MIC of 5 mg mL^−1^, which was a better outcome compared to some previous samples. It also inhibited *S. enterica* at an MIC of 2.5 mg mL^−1^ and showed moderate activity against *S. aureus* with an MIC of 5 mg mL^−1^. No effective inhibition was observed for the other bacteria, with MICs greater than 10 mg mL^−1^. None of the samples tested exhibited the capacity to kill the bacteria, as evidenced by the Minimum Fungicidal Concentration (MFC) values for all samples being greater than 10 mg mL^−1^. When examining the antibacterial activity of various accessions from Table 4, it becomes evident that there is a notable variation in their effectiveness against different bacteria. The data demonstrate a range of Minimum Inhibitory Concentration (MIC) and Minimum Bactericidal Concentration (MBC) across the samples when tested against Gram-negative and Gram-positive bacteria.

*S. aureus* was particularly sensitive to the samples, often being inhibited at lower MIC values, such as 2.5 mg mL^−1^. This was a consistent finding across several accessions, suggesting these accessions could be potential candidates for therapeutic use against infections caused by this bacterium. Similarly, *S. enterica* displayed susceptibility, although not as pronounced as *S. aureus*, with several samples achieving inhibition at MIC values of either 2.5 mg mL^−1^ or 5 mg mL^−1^. *E. coli* showed a moderate level of susceptibility, with effective samples generally having MIC values around 10 mg mL^−1^.

Some samples stood out for their effectiveness. For instance, accession BPGV02906 showed notable activity against *S. aureus* with an MIC of 2.5 mg mL^−1^ and moderate activity against *E. coli* and *S. enterica*. Similarly, accessions BPGV16127 and BPGV07480 were particularly effective against *S. aureus* and *S. enterica*, with low MIC values indicating strong inhibitory effects. Accession BPGV03990 exhibited good activity against *E. coli*, *S. enterica*, and *S. aureus*, and BPGV12239 showed better outcomes against *E. coli* and excellent inhibition of *S. enterica*.

However, not all accessions were effective. A significant number of them showed no effective inhibition against bacteria such as *E. cloacae*, *P. aeruginosa*, *Y. enterocolitica*, *B. cereus*, and *L. monocytogenes*, with MIC values exceeding 10 mg mL^−1^. This suggests that these bacteria are more resistant to the samples tested.

Overall, the study indicates that while some accessions show potential as antimicrobial agents, particularly against *S. aureus* and to some extent against *S. enterica* and *E. coli*, there is a lack of bactericidal activity at the concentrations tested. The variation in the efficacy of the samples, with some showing broad-spectrum activity and others being effective against a more limited range of bacteria, emphasizes the need for further research. This research should aim to enhance the efficacy and expand the spectrum of these antimicrobial agents, as well as to understand the mechanisms of resistance in the less susceptible bacteria. Comparing these findings with the study on *B. rapa* extracts by [55], which demonstrated positive inhibitory effects against *C. albicans*, *P. aeruginosa*, and *B. subtilis*, it can be inferred that extracts from *B. rapa* possess a different antibacterial activity spectrum. For example, while the *B. rapa* extracts were effective against *B. subtilis*, the extracts in our study did not show significant activity against *B. cereus*, closely related to *B. subtilis*. Furthermore, the MIC values for effective inhibition in the *B. rapa* study (12.5–25 mg mL^−1^) were higher compared to the lower MIC values observed in some of our extracts (2.5–10 mg mL^−1^).

Another study by [56] concluded that the compound 2-phenylethyl isothiocyanate found in *B. rapa* samples showed significant activity against *Vibrio parahaemolyticus*, *S. aureus*, and *B. cereus*, particularly effective against *V. parahaemolyticus* with an MIC of 100 μg mL^−1^. This indicates the potential of specific plant-derived compounds as highly effective antimicrobial agents. Additionally, the inhibition of *Helicobacter pylori* by turnip root extracts suggests a potential role in treating *H. pylori* infections.

Both studies provide valuable insights into the potential of plant extracts as antimicrobial agents. However, their effectiveness varies depending on the type of extract and the microbial strain. The variation in MIC values among different extracts and organisms highlights the complexity of developing plant-based antimicrobial treatments and the need for detailed, strain-specific research. These results suggest that the efficacy of samples varies significantly across different bacterial strains. Some samples show potential for therapeutic use, especially against *S. aureus*, but none seem to possess strong bactericidal properties at the tested concentrations. This information is crucial for understanding the potential use of these accessions as antimicrobial agents and may guide further research and development.

### 3.4. Principal Component Analysis (PCA) to Assess Overall Variation

Fourteen GLS were identified among the group of 27 *B. rapa* accessions from traditional varieties (landraces).

The main compounds identified in the sampling of Portuguese landraces from *B. rapa* were GST, GBNISOM, NGBS, and 4-OHGBSisom.

The box plot graph (Figure 2) presents the variation between the average values of accessions by region of origin. It is evident that there is greater variability in all the GLS compounds identified in the accessions collected in Aveiro, except for 4-OHGBS and NGBS compounds. The longer length of the boxes for GNA and GST compounds suggests greater variability in the data of the accessions collected in six regions. The boxes for 4-OHGBS, GBN-isomer, GER, and GNL compounds in the Aveiro and Braga regions overlap, suggesting no significant difference in central tendency between those regions. The same trend was observed for 4-OHGBS and its isomer, GBN, GER, and PRO collected in Braga and Bragança. Non-overlapping boxes for the other compounds may indicate a potential difference in average values between regions.

It is important to note that all areas of origin are represented in the GLS analysis. In order to improve the identification of GLS profiles in traditional *B. rapa* varieties, future evaluations should include more samples from Vila Real, Leiria, Santarém, and Portalegre.

A Principal Component Analysis (PCA) biplot (Figure 3) was performed to match the interactions between the variability in GLS profiles within the species of the *B. rapa* collection with the nine regions where the turnips were harvested.

The data points are represented in the space defined by the first two principal components (PC1 = 34.44% and PC2 = 21.92%) and explain 56.36% of the total variance. The vectors associated with the original variables are also plotted in the same space. The vectors’ angle and length indicate the correlation between variables and the importance of each variable in defining the principal components.

Statistically, it is preferable for the percentage of variance explained by the two principal components to be greater than 70%. The analysis revealed that 56% of the variance was explained. However, this percentage may be an underestimation due to the lack of contribution from some accessions. 

Figure 3 shows that, within principal component 2, the vectors for GOB, PRO, GBN, and GER point in similar directions, indicating positive correlations. Conversely, the vectors for GST, GNA, GBS, NBS, GBA, 4-OHGBS, and isomer point in different directions, suggesting negative correlations. The length of the vectors reflects the variables’ importance in explaining the data’s variance, with longer vectors representing variables that contribute more to the principal components. The analysis indicates that certain GLS profiles seem to be more prevalent in certain regions where the turnips were harvested.

There is a wider GLS profile associated with the inland areas with a continental influence Bragança, Viseu, Guarda, since accessions from these locations tend to spread over both principal components. On the other hand, accessions from Aveiro tend to accumulate on the bottom part of the PCA, below any other accession, most likely revealing a greater effect of GBA and NBGS, which was already noticeable from Figure 2. Vila Real also presented an interesting location in the PCA, occupying a unique space outside the main cluster. Although Vila Real is only represented by one single accession (BPGV07480), this result could indicate a regional effect on GLS profiling, namely for GNL, which was at least 3-fold when compared with any other accession, and also very low values for GST.

Accession BPGV05888 belongs to the taxonomic classification of *B. rapa sylvestris*, commonly known as the wild form of B. campestris, and also referred to in the literature as the wild or weedy form of *B. rapa* [11,12]. Similar to other studies, this taxon differs from other *B. rapa* accessions in its GLS profile, despite being represented by a single accession. In the study conducted by [57], the authors identified the compounds GNA and GBN for the first time. In this study, we found that the main compounds in the accession’s GLS profile were GNA and GST.

Accessions BPGV06207 and BPGV12239 have a distinct GLS profile. The first is a landrace cultivated for sprouts, although it produces edible roots. The second landrace was designated by the farmer as *B. rapa*, but morphological taxonomy studies indicate that it is *B. napus* napobrassica and the GLS profile obtained may reflect this fact.

Hierarchical clustering was used to group similar accessions based on their GLS profiles into clusters. The resulting clusters are presented in the dendrogram’s hierarchical tree-like structure (Figure 4). The clusters appear to be influenced by the year of characterisation, with more cohesive groups forming within each agricultural year. However, they are not grouped together in the dendrogram.

There is variability in the GLS profiles between *B. rapa* landraces, with different profiles observed between *B. rapa* accessions from the same area, characterised in the same year and on the same plot.

The soil and climate conditions influenced the GLS profiles identified and detected, regardless of the organ being analysed (in this case turnips), as expected and considered by other authors [29,36,37].

### 3.5. Conclusions

The turnip breeding program could be useful for a deeper understanding of these crops for both nutritional quality and bioaccessibility, focusing on turnips with specific characteristics, namely GLS and phenolic compounds, as well as, antioxidant and antimicrobial activities, indicating varieties that contribute health benefits and adapted to Mediterranean conditions, thereby contributing to the diversification of, traditional or not, Brassica vegetable production systems, and may have disease-fighting properties potential, associated with the consumption of cruciferous. 

The results from HRMS allowed the identification of a diverse range of chemical constituents from *Brassica rapa* and *Brassica napus* landraces conserved in ex situ collections in Portugal. These compounds include 14 GLS compounds and 7 hydroxyciannamic acids and their derivatives. PRO was the major compound identified in the turnip extracts of accession BPGV06207 and has been related to antithyroid effects. GBS was identified in the extracts of accession BPGV13051 and has been associated with anticancer and anti-inflammatory properties. These compounds could be responsible for the observed biological activities and may have potential health benefits.

There was observed some variability in the antioxidant activities among different Brassica accessions. Some landraces (BPGV06207 accession) exhibited significant antioxidant properties, while others (BPGV12736 and BPGV05049 accessions) have shown relatively lower activity in FRAP and DPPH▪ assays, respectively.

The antibacterial activity has shown variability among the 16 turnip accessions. A total of 11 accessions exhibited significant activity against two bacteria (*S. enterica* and *S. aureus*) with MIC values (2.5 mg mL^−1^), two accessions (BPGV12239 and BPGV02906) revealed also inhibition against *E. coli*) with MIC values of about 5 mg mL^−1^, and only one accession (BPGV05875) inhibited *Y. enterocolitica* with 5 mg mL^−1^ MIC value. These results suggest that these accessions could be potential candidates for therapeutic use against infections caused by the indicated bacteria.

The variation in the efficacy of the accessions, with some showing broad-spectrum activity and others being effective against a more limited range of bacteria, emphasizes the need for further research. This research aimed to enhance the efficacy and expand the spectrum of these antimicrobial agents, as well as to understand the mechanisms of resistance in the less susceptible bacteria.

The results suggest that consuming *Brassica rapa* and *Brassica napus* varieties, particularly those grown in Portuguese environments, could potentially contribute to human health due to their bioactive properties.

The findings have implications for agricultural practices, crop diversification, and industry stakeholders involved in *Brassica* production and product development. Understanding the chemical and biological characteristics of Brassica varieties stands for breeding strategies aimed at enhancing desirable traits for both agricultural productivity and human health. Similarly, the development of Brassica-based products with optimised bioactive profiles could cater to consumer demand for functional foods and natural health supplements.

## Figures and Tables

**Figure 1 foods-13-01164-f001:**
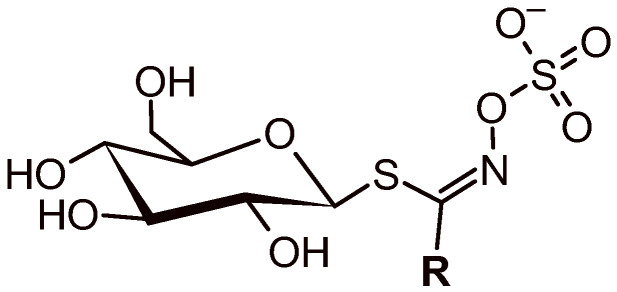
General structure of a deprotonated molecule of a GLS.

**Figure 2 foods-13-01164-f002:**
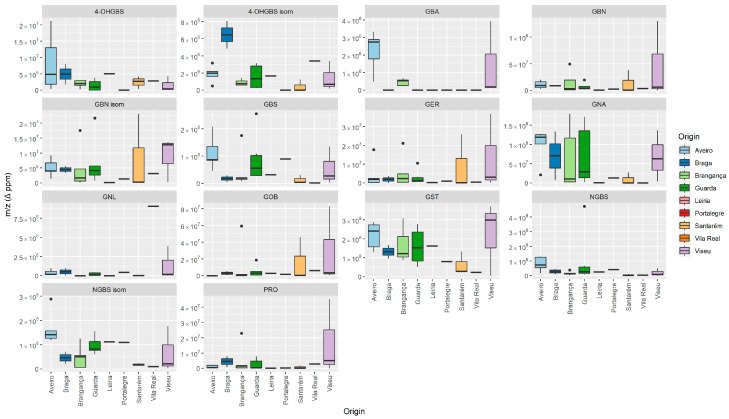
Variation between the average values (*m*/*z* (Δ ppm)) of the accessions by region of origin.

**Figure 3 foods-13-01164-f003:**
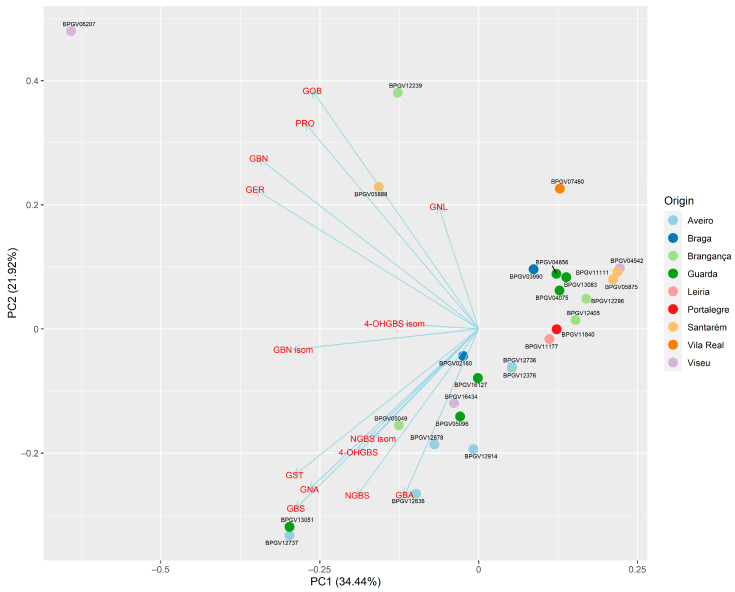
Principal component analysis biplot based on average values matrix of GLS identifies 27 accessions of *B. rapa*. The different colours represent the different districts, and each district’s accessions are marked by the same colours.

**Figure 4 foods-13-01164-f004:**
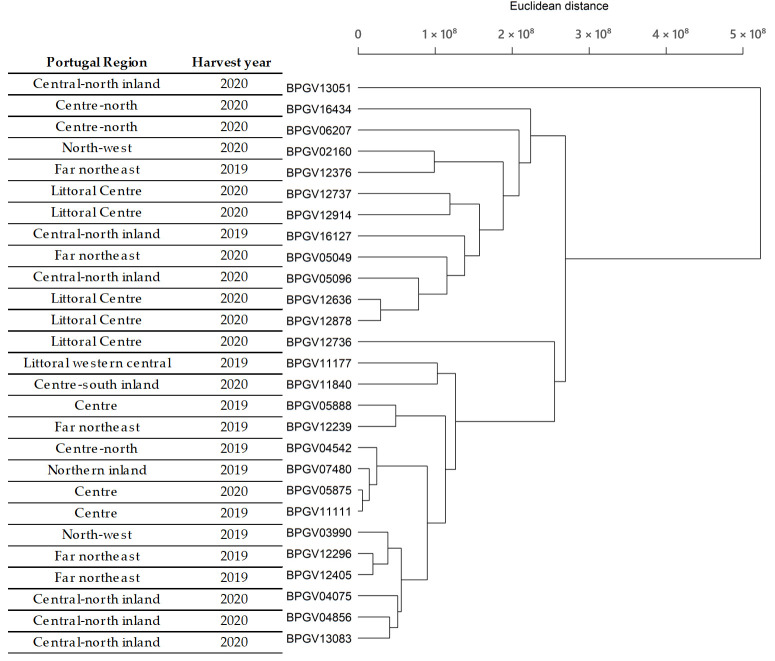
Hierarchical clustering of the 27 accessions based on their LC-MS profile (r = 0.915).

**Table 1 foods-13-01164-t001:** Identification of the turnip and rapeseed accessions evaluated, their origin, sample, and harvest year.

Accession	County Origin	Portugal Region	Species	Sample Analized	Harvest Year
BPGV03990	Braga	North-west	*Brassica rapa*	turnip	2019
BPGV12376	Bragança	Far northeast	*Brassica rapa*	turnip	2019
BPGV12405	Bragança	Far northeast	*Brassica rapa*	turnip	2019
BPGV12239	Bragança	Far northeast	*Brassica rapa*	turnip	2019
BPGV12296	Bragança	Far northeast	*Brassica rapa*	turnip	2019
BPGV16127	Guarda	Central-north inland	*Brassica rapa*	turnip	2019
BPGV11177	Leiria	Littoral western central	*Brassica rapa*	turnip	2019
BPGV05875	Santarém	Centre	*Brassica rapa*	turnip	2019
BPGV11111	Santarém	Centre	*Brassica rapa*	turnip	2019
BPGV04542	Viseu	Centre-north	*Brassica rapa*	turnip	2019
BPGV07301	Vila Real	Northern inland	*Brassica rapa*	turnip	2019
BPGV07480	Vila real	Northern inland	*Brassica rapa*	turnip	2019
BPGV06987A	Bragança	Far northeast	*Brassica napus* var. napobrassica (L.) Rchb.	turnip	2019
BPGV06987B	Bragança	Far northeast	*Brassica napus* var. napobrassica (L.) Rchb.	turnip	2019
BPGV02906	Vila real	Northern inland	*Brassica napus* var. napobrassica (L.) Rchb.	turnip	2019
BPGV05888	Santarém	Centre	*Brassica rapa* sylvestris	turnip	2019
BPGV12636	Aveiro	Littoral Centre	*Brassica rapa*	turnip	2020
BPGV12736	Aveiro	Littoral Centre	*Brassica rapa*	turnip	2020
BPGV12737	Aveiro	Littoral Centre	*Brassica rapa*	turnip	2020
BPGV12878	Aveiro	Littoral Centre	*Brassica rapa*	turnip	2020
BPGV12914	Aveiro	Littoral Centre	*Brassica rapa*	turnip	2020
BPGV05049	Bragança	Far northeast	*Brassica rapa*	turnip	2020
BPGV13051	Guarda	Central-north inland	*Brassica rapa*	turnip	2020
BPGV13083	Guarda	Central-north inland	*Brassica rapa*	turnip	2020
BPGV05096	Guarda	Central-north inland	*Brassica rapa*	turnip	2020
BPGV11840	Portalegre	Centre-south inland	*Brassica rapa*	turnip	2020
BPGV06207	Viseu	Centre-north	*Brassica rapa*	turnip	2020
BPGV16434	Viseu	Centre-north	*Brassica rapa*	turnip	2020
BPGV04856	Guarda	Central-north inland	*Brassica rapa* subsp. rapa	turnip	2020
BPGV04075	Guarda	Central-north inland	*Brassica rapa* subsp. rapa	turnip	2020
BPGV02160	Braga	North-west	*Brassica rapa* subsp. rapa	turnip	2020
BPGV02578	Braga	North-west	*Brassica napus*	seeds	2021
BPGV04238	Braganca	Far northeast	*Brassica napus*	seeds	2021
BPGV02636	Viana Castelo	Littoral North-west	*Brassica napus*	seeds	2021
BPGV02740	Viana Castelo	Littoral North-west	*Brassica napus*	seeds	2021
BPGV02745	Viana Castelo	Littoral North-west	*Brassica napus*	seeds	2021
BPGV02707	Vila Real	Northern inland	*Brassica napus*	seeds	2021

The identification of the accessions follows the Nomenclature and genomic relationships of cultivated Brassica species and related genera from the [38].

**Table 2 foods-13-01164-t002:** UHPLC-ESI(-)-HRMS/MS characterisation of the main compounds present in the turnip methanol extracts.

Semisystematic Name (Glucosinolate Group)	Trivial Name (Abbreviation)	T*_R_* (min)	Ionic Formula	[M − H]^−^ (*m*/*z* (Δ ppm)	MS/MS [*m*/*z* (Δ ppm) (Attribution)]
2(*R*)-2-Hydroxyl but- -3-enyl (hydroxyalkenyl)	Progoitrin (PRO)	1.38	[C_11_H_19_N_1_O_10_S_2_]^−^	388.0380; −0.7	308.0805; −2.4; (C_11_H_18_N_1_O_7_S_1_)^−^ 301.0124; −0.5; (C_8_H_13_O_8_S_2_)^−^ 274.9904; −1.0; (C_6_H_11_O_8_S_2_)^−^ 259.0131; −0.6; (C_6_H_11_O_9_S_1_)^−^ 241.0029; −2.3; (C_6_H_9_O_8_S_1_)^−^ 195.0334; −0.6; (C_6_H_11_O_5_S_1_)^−^ **194.9794; −1.3; (C_6_H_7_O_4_S_2_)^−^*^F2b^** 179.0560; −0.7; (C_5_H_11_O_6_)^−^ **146.0278; +1.9; (C_5_H_8_N_1_O_2_S_1_)^−^*^F1^** 135.9706; +3.1; (C_2_H_2_N_1_O_4_S_1_)^−^
2-Hydroxypent- -4-enyl (hydroxyalkenyl)	Gluco- napoleiferin (GNL)	1.45	[C_12_H_20_N_1_O_10_S_2_]^−^	402.0543; −2.3	322.0950; −2.4; (C_12_H_20_N_1_O_7_S_1_)^−^ 274.9902; −1.9; (C_6_H_11_O_8_S_2_)^−^ 259.0131; −2.1; (C_6_H_11_O_9_S_1_)^−^ **208.9952; −2.2; (C_6_H_9_O_4_S_2_)^−^*^F2b^** 195.0333; −0.9; (C_6_H_11_O_5_S_1_)^−^ **160.0437; +0.2; (C_6_H_10_N_1_O_2_S)^−^*^F1^** 135.9713; −2.0; (C_2_H_2_N_1_O_4_S)^−^
But-3-enyl (alkenyl)	Gluconapin (GNA)	1.47	[C_11_H_18_N_1_O_9_S_2_]^−^	372.0432; −1.0	292.0861; −0.4; (C_11_H_18_N_1_O_6_S)^−^ 274.9905; −1.5; (C_6_H_11_O_8_S_2_)^−^ 259.0133; −1.3; (C_6_H_11_O_9_S_1_)^−^ 241.0025; −0.5; (C_6_H_9_O_8_S_1_)^−^ 195.0333; −0.9; (C_6_H_11_O_5_S_1_)^−^ **178.9834; −4.4; (C_5_H_7_O_3_S_2_)^−^*^F2b^** **130.0334; −1.2; (C_5_H_8_NOS)^−^*^F1^**
4-Hydroxyindol- -3-ylmethyl (indole)	4-Hydroxy- glucobrassicin (4-OHGBS)	1.55	[C_16_H_19_N_2_O_10_S_2_]^−^	463.0491; −0.9	383.0921; −0.6; (C_16_H_19_N_2_O_7_S)^−^ 274.9903; −0.4; (C_6_H_11_O_8_S_2_)^−^ **267.0081; −0.2; (C_10_H_7_N_2_O_5_S)^−^ *^F2a^** 259.0128; −0.4; (C_6_H_11_O_9_S_1_)^−^ **221.0390; −0.3; (C_10_H_9_N_2_O_2_S_1_)^−^*^F1^**
Pent-4-enyl (alkenyl)	Gluco- brassicanapin (GBN)	2.20	[C_12_H_20_N_1_O_9_S_2_]^−^	386.0583; −0.6	306.1011; −1.1; (C_12_H_20_N_1_O_6_S)^−^ 274.9900; −0.3; (C_6_H_11_O_8_S_2_)^−^ 259.0131; −0.6; (C_6_H_11_O_9_S_1_)^−^ 241.0021; −1.1; (C_6_H_9_O_8_S_1_)^−^ 195.0332; −0.4; (C_6_H_11_O_5_S)^−^ **190.0187; −3.8; (C_6_H_8_NO_4_S_1_)^−^*^F2a^** **144.0484; −2.9; (C_6_H_10_NOS_1_)^−^*^F1^**
Hydroxyindol- -3-ylmethyl (indole)	Hydroxy- glucobrassicin isomer (OHGBS isomer)	3.02	[C_16_H_19_N_2_O_10_S_2_]^−^	463.0493; −1.1	383.0917; −0.4; (C_16_H_19_N_2_O_7_S)^−^ 274.9902; −0.3; (C_6_H_11_O_8_S_2_)^−^ **267.0092; −4.2; (C_10_H_7_N_2_O_5_S)^−^*^F2a^** 259.0131; −0.6; (C_6_H_11_O_9_S_1_)^−^ **221.0391; −0.4; (C_10_H_9_N_2_O_4_S_1_)^−^*^F1^** 195.0339; −4.4; (C_6_H_11_O_5_S)^−^
4(Methylsulphanyl)- -butyl (sulphur containing)	Glucoerucin (GER)	3.21	[C_12_H_22_N_1_O_9_S_3_]^−^	420.0467; −1.2	340.0913; −5.4; (C_12_H_22_N_1_O_6_S_2_)^−^ 331.0716; −1.4; (C_11_H_22_N_1_O_6_S_2_)^−^ 274.9900; −1.5; (C_6_H_11_O_8_S_2_)^−^ 259.0140; −4.3; (C_6_H_11_O_9_S_1_)^−^ **226.9884; −3.5; (C_6_H_11_O_3_S_3_)^−^*^F2b^** 195.0340; −4.9; (C_6_H_11_O_5_S)^−^ **178.0366; −0.1; (C_6_H_12_N_1_OS_2_)^−^*^F1^** 130.0334; −1.2; (C_5_H_8_NOS)^−^
Pentenyl (alkenyl)	Gluco- brassicanapin isomer (GBN isomer)	3.38	[C_12_H_20_N_1_O_9_S_2_]^−^	386.0595; −2.6	306.1024; −2.5; (C_12_H_20_N_1_O_6_S)^−^ 274.9905; −2.6; (C_6_H_11_O_8_S_2_)^−^ 259.0133; −1.3; (C_6_H_11_O_9_S_1_)^−^ 241.0035; −2.0; (C_6_H_9_O_8_S_1_)^−^ 195.0339; −2.5; (C_6_H_11_O_5_S)^−^ **190.0187; −3.8; (C_6_H_8_NO_4_S)^−^*^F2a^** **144.0487; −0.9; (C_6_H_10_NOS)^−^*^F1^**
Indol-3-ylmethyl (indole)	Glucobrassicin (GBS)	3.40	[C_16_H_19_N_2_O_9_S_2_]^−^	447.0547; −2.0	367.0962; −2.0; (C_16_H_19_N_2_O_6_S_1_)^−^ 274.9909; −3.1; (C_6_H_11_O_8_S_2_)^−^ 259.0138; −3.4; (C_6_H_11_O_9_S_1_)^−^ **253.9961; −3.9; (C_10_H_8_N1O_3_S_2_)^−^*^F2b^** **205.0440; −0.5; (C_10_H_9_N_2_OS)^−^*^F1^** 172.0223; −2.1; (C_10_H_6_N_1_S_1_)^−^
2(*R*)-Hydroxy- -2-phenylethyl (aromatic)	Glucobarbarin (GBA)	3.70	[C_15_H_20_N_1_O_10_S_2_]^−^	438.0534; −3.7	358.0996; −7.4; (C_15_H_20_N_1_O_7_S_1_)^−^ 274.9909; −3.1; (C_6_H_11_O_8_S_2_)^−^ 259.0140; −4.1; (C_6_H_11_O_9_S_1_)^−^ **244.9941; −2.8; (C_9_H_9_O_4_S_2_)^−^*^F2b^** **196.0442; −2.0; (C_9_H_10_NO_2_S_1_)^−^*^F1^** 172.0223; −2.1; (C_10_H_6_N_1_S_1_)^−^
5-(Methylsulfanyl)- -pentyl (sulfur containing)	Glucoberteroin (GOB)	4.46	[C_13_H_24_N_1_O_9_S_3_]^−^	434.0626; −1.7	354.1060; −2.8; (C_13_H_24_N_1_O_6_S_2_)^−^ 274.9903; −0.7; (C_6_H_11_O_8_S_2_)^−^ 259.0140; −1.8; (C_6_H_11_O_9_S_1_)^−^ **241.0035; −1.1; (C_7_H_13_O_3_S_3_)^−^*^F2b^** 195.0335; −1.3; (C_6_H_11_O_5_S_1_)^−^ **192.0524; −1.3; (C_7_H_14_N_1_O_1_S_2_)^−^*^F1^**
2-Phenylethyl (aromatic)	Gluconasturtiin (GST)	4.47	[C_15_H_20_N_1_O_9_S_2_]^−^	422.0593; −2.0	342.1026; −2.8; (C_15_H_20_N_1_O_6_S_1_)^−^ 274.9912; −3.9; (C_6_H_11_O_8_S_2_)^−^ 259.0295; −3.6; (C_6_H_11_O_9_S_1_)^−^ 241.0032; −3.5; (C_6_H_9_O_8_S_1_)^−^ **229.0006; −3.1; (C_9_H_9_O_3_S_2_)^−^*^F2b^** **226.0184; −1.8; (C_9_H_8_N_1_O_4_S_1_)^−^*^F2a^** **180.0485; −1.8; (C_9_H_10_N_1_O_1_S_1_)^−^*^F1^**
4-Methoxyindol- -3-ylmethyl (indole)	4-methoxy- glucobrassicin (NGBS)	4.78	[C_17_H_21_N_2_O_10_S_2_]^−^	477.0655; −2.5	**284.0062; −2.0; (C_11_H_10_N_1_O_4_S_2_)^−^*^F2b^** 274.9911; −3.9; (C_6_H_11_O_8_S_2_)^−^ 259.0138; −0.9; (C_6_H_11_O_9_S_1_)^−^ **235.0560; −5.8; (C_11_H_11_N_2_O_2_S_1_)^−^*^F1^** 205.0441; −0.2; (C_10_H_9_N_2_O_1_S_1_)^−^ 203.0286; −0.5; (C_10_H_7_N_2_O_1_S_1_)^−^
1-Methoxyindol- -3-ylmethyl (indole)	1-methoxy- -glucobrassicin (NGBS isomer)	5.62	[C_17_H_21_N_2_O_10_S_2_]^−^	477.0664; −3.7	446.0480; −4.6; (C_16_H_18_N_2_O_9_S_2_)^−•^ 274.9900; −0.9; (C_6_H_11_O_8_S_2_)^−^ 259.0136; −2.6; (C_6_H_11_O_9_S_1_)^−^ **235.0559; −5.2; (C_11_H_11_N_2_O_2_S_1_)^−^*^F1^** 205.0437; −1.9; (C_10_H_9_N_2_O_1_S_1_)^−^ 195.0344; −6.0; (C_6_H_11_O_5_S)^−^
Hydroxyciannamic acids					
	Sucrose	1.13	[C_12_H_21_O_11_]^−^	341.1091; −0.5	179.0562; −0.7; (C_8_H_11_O_6_)^−^
	Gentisoyl glucoside	1.93	[C_13_H_15_O_9_]^−^	315.0726; −1.5	153.02187; +4,7; (C_7_H_5_O_4_)^−^
	1-*O*-Sinapolyl- glucose	4.80	[C_17_H_21_O_10_]^−^	385.1139; −0.2	205.0510; −1.8; (C_11_H_9_O_4_)^−^ 179.0723; −5.3; (C_10_H_11_O_3_)^−^ 164.0480; −0.4; (C_9_H_8_O_3_)^−•^
	Sinapic acid	6.17	[C_11_H_11_O_5_]^−^	223.0612; −1.9	208.0375; −0.7; (C_10_H_8_O_5_)^−•^ 179.0708; −7.7; (C_10_H_11_O_3_)^−^ 164.0469; −5.7; (C_9_H_8_O_3_)^−•^
	1,2-Disinapoyl-gentiobiose	6.73	[C_34_H_41_O_19_]^−^	753.2252; −0.5	529.1571; −0.8; (C_23_H_29_O_14_)^−^ 223.0615; −1.3; (C_11_H_11_O_5_)^−^ 205.0510; −1.4; (C_11_H_9_O_4_)^−^
	1-Feruloyl- -1-sinapoyl- gentiobiose	6.88	[C_33_H_39_O_18_]^−^	723.2145; −0.4	449.1471; −2.7; (C_22_H_27_O_13_)^−^ 223.0615; −3.6; (C_11_H_11_O_5_)^−^ 205.0517; −5.0; (C_11_H_9_O_4_)^−^ 175.0463; −1.6; (C_10_H_7_O_3_)^−^
	1,2,2′-Trisinapoyl- Gentiobiose	7.46	[C_45_H_51_O_23_]^−^	959.2855; −2.5	735.2155; −1.0; (C_34_H_39_O_18_)^−^ 529.1578; −2.8; (C_23_H_29_O_14_)^−^ 223.0622; −4.7; (C_11_H_11_O_5_)^−^ 205.0505; −0.5; (C_11_H_9_O_4_)^−^
	1,2′-Disinapoyl-2- -feruloyl- Gentiobiose	8.15	[C_44_H_49_O_22_]^−^	929.2721; −3.8	705.2057; −2.9; (C_33_H_37_O_17_)^−^ 529.1575; −2.3; (C_23_H_29_O_14_)^−^ 205.0518; −5.5; (C_11_H_9_O_4_)^−^

* Symbols F1 and F2a or F2b indicate fragmentation paths which are specific to each GLS molecular structure (differentiated by the R functional group).

**Table 3 foods-13-01164-t003:** Antioxidant activity and total phenolic compounds for *B. rapa* and *B. napus* turnip extracts.

Accession	TPC (mg GAE·g^−1^) DW	FRAP (µmolFe^2+^·g^−1^) DW	DPPH (mmolTE·g^−1^) DW
BPGV03990	7.85 ± 0.39 ^a,b,c,d,e,f^	79.4 ± 2.99 ^a,b^	157.59 ± 0.16 ^b,c^
BPGV12376	9.66 ± 0.11 ^c,d,e,f,g,h,i^	73.50 ± 4.65 ^a,b,c,d^	208.28 ± 8.77 ^b,c,d,e^
BPGV12405	11.91 ± 1.19 ^h,i,j,l^	102.96 ± 3.77 ^a,b,c^	177.96 ± 1.61 ^b,c,d,e^
BPGV12239	8.66 ± 2.60 ^a,b,c,d,e,f,g,h^	51.08 ± 11.93 ^a,b,c,d^	187.11 ± 2.01 ^b,c,d,e^
BPGV12296	10.47 ± 0.77 ^f,g,h,i,j^	88.83 ± 12.23 ^a,b,c^	212.60 ± 0.08 ^b,c,d,e^
BPGV16127	10.05 ± 0.54 ^d,e,f,g,h,i^	97.64 ± 3.32 ^a^	202.36 ± 1.69 ^b,c,d,e^
BPGV11177	8.17 ± 0.06 ^a,b,c,d,e,f,g^	63.61 ± 6.85 ^a^	186.72 ± 4.02 ^b,c,d,e^
BPGV05875	12.10 ± 0.18 ^h,i,j,l^	106.41 ± 0.29 ^a,b,c,d^	188.48 ± 0.08 ^b,c,d,e^
BPGV11111	11.76 ± 2.52 ^h,i,j,l^	74.80 ± 21.11 ^a^	179.15± 9.41 ^b,c,d,e^
BPGV04542	9.13 ± 0.45 ^c,d,e,f,g,h,i^	68.48 ± 2.79 ^a^	184.10 ± 6.44 ^b,c,d,e^
BPGV07301	5.38 ± 0.97 ^a^	59.70 ± 16.08 ^a,b,c,d^	230.40 ± 7.40 ^e^
BPGV07480	5.61 ± 0.88 ^a,b^	51.03 ± 1.50 ^a,b,c^	194.28 ± 30.97 ^b,c,d,e^
BPGV06987A	6.89 ± 0.22 ^a,b,c,d^	60.68 ± 4.57 ^a,b,c,d^	202.36 ± 1.69 ^b,c,d,e^
BPGV06987B	10.52 ± 0.49 ^i,j^	72.00 ± 5.70 ^a,b,c^	195.65 ± 5.07 ^b,c,d,e^
BPGV02906	6.72 ± 0.67 ^a,b,c,d^	55.17 ± 4.94 ^a,b,c^	213.91 ± 3.38 ^b,c,d,e^
BPGV05888	7.01 ± 0.55 ^a,b,c,d,e^	59.74 ± 8.60 ^a,b,c^	196.59 ± 6.80 ^b,c,d,e^
BPGV12636	7.91 ± 0.51 ^a,b,c,d,e,f^	64.43 ± 7.80 ^a,b,c,d,e^	210.08 ± 7.56 ^b,c,d,e^
BPGV12736	6.41 ± 0.09 ^a,b,c^	44.01 ± 10.69 ^a,b,c^	156.91 ± 2.95 ^b^
BPGV12737	13.72 ± 0.71 ^j,l,m^	123.92 ± 14.21 ^e^	228.3 ± 0.16 ^e^
BPGV12878	11.73 ± 0.07 ^h,i,j,l^	85.04 ± 2.73 ^c,d,e^	225.44 ± 1.50 ^e^
BPGV12914	18.00 ± 0.64 ^n^	127.97 ± 0.15 ^e^	218.72 ± 2.52 ^d,e^
BPGV05049	12.58 ± 0.00 ^i,j,l^	69.31 ± 7.01 ^a,b,c,d^	94.00 ± 17.70 ^a^
BPGV13051	8.75 ± 0.42 ^a,b,c,d,e,f,g,h^	65.94 ± 22.57 ^a,b,c,d^	193.16 ± 32.34 ^b,c,d,e^
BPGV13083	15.12 ± 0.03 ^l,m,n^	124.06 ± 38.57 ^e^	184.67 ± 51.43 ^b,c,d,e^
BPGV05096	9.79 ± 0.29 ^c,d,e,f,g,h,i^	83.75 ± 5.55 ^b,c,d,e^	216.71 ± 6.22 ^d,e^
BPGV11840	16.90 ± 0.83 ^m,n^	104.37 ± 0.42 ^d,e^	210.23 ± 3.16 ^b,c,d,e^
BPGV16434	7.09 ± 0.46 ^a,b,c,d,e,f^	56.92 ± 20.44 ^a,b,c,d^	168.89 ± 1.45 ^b,c,d^
BPGV06207	16.92 ± 0.66 ^m,n^	130.83 ± 24.98 ^e^	232.53 ± 1.56 ^e^
BPGV04856	8.99 ± 0,45 ^h,i,j,l^	63.61 ± 5.34 ^a,b,c,d,e^	190.51 ± 11.10 ^b,c,d,e^
BPGV04075	12.06 ± 0.58 ^b,c,d,e,f,g,h^	70.51 ± 1.21 ^a,b,c,d^	211.63 ± 9.01 ^b,c,d,e^
BPGV02160	11.43 ± 0.44 ^g,h,i,j^	81.92 ± 6.55 ^b,c,d,e^	221.07 ± 1.13 ^d,e^

Values are expressed as average ± standard deviation of two parallel experiments. DW: dry weight GAE: Gallic acid equivalents; TE: Trolox equivalents; Different letters in the same column indicate significant differences (*p* < 0.05). Statistical evaluation was performed using ANOVA test.

**Table 4 foods-13-01164-t004:** Antioxidant activity and total phenolic compounds for *B. napus* turnip seed extracts.

Accession	TPC (mg GAE·g^−1^) DW	FRAP (mmolFe^2+^·g^−1^) DW	DPPH (mmol·TE g^−1^) DW
BPGV02578	27.09 ± 2.69 ^a^	3.18 ± 0.15 ^a^	330.88 ± 40.58 ^a^
BPGV04238	24.90 ± 5.16 ^a^	3.01 ± 0.35 ^a^	280.38 ± 65.11 ^a^
BPGV02636	23.04 ± 0.94 ^a^	2.74 ± 0.16 ^a^	246.44 ± 2.59 ^a^
BPGV02740	27.39 ± 0.20 ^a^	3.12± 0.50 ^a^	314.81 ± 12.71 ^a^
BPGV02745	25.12 ± 0.37 ^a^	2.89 ± 0.01 ^a^	296.73 ± 8.55 ^a^
BPGV02707	24.40 ± 0.72 ^a^	2.99 ± 0.66 ^a^	286.37 ± 8.36 ^a^

Values are expressed as average ± standard deviation of two parallel experiments. DW: dry weight GAE: Gallic acid equivalents; TE: Trolox equivalents. Different letters in the same column indicate significant differences (*p* < 0.05). Statistical evaluation was performed using ANOVA test.

**Table 5 foods-13-01164-t005:** Antibacterial activity (MIC and MBC; mg mL^−1^) of turnip extracts.

		Gram-Negative Bacteria					Gram-Positive Bacteria		
Sample		*Enterobacter Cloacae*	*Escherichia coli*	*Pseudomonas aeruginosa*	*Salmonella enter* *ica*	*Yersinia enterocolitica*	*Bacillus cereus*	*Listeria monocytogenes*	*Staphylococcus aureus*
BPGV03990	MIC	>10	10	>10	2.5	>10	>10	>10	2.5
	MBC	>10	>10	>10	>10	>10	>10	>10	>10
BPGV12239	MIC	>10	5	>10	2.5	>10	>10	>10	5
	MBC	>10	>10	>10	>10	>10	>10	>10	>10
BPGV12296	MIC	>10	10	>10	5	10	>10	>10	5
	MBC	>10	>10	>10	>10	>10	>10	>10	>10
BPGV12376	MIC	>10	10	>10	5	10	>10	>10	5
	MBC	>10	>10	>10	>10	>10	>10	>10	>10
BPGV12405	MIC	>10	10	>10	5	>10	>10	>10	2.5
	MBC	>10	>10	>10	>10	>10	>10	>10	>10
BPGV16127	MIC	>10	10	>10	5	10	>10	>10	2.5
	MBC	>10	>10	>10	>10	>10	>10	>10	>10
BPGV11177	MIC	>10	10	>10	5	10	>10	>10	2.5
	MBC	>10	>10	>10	>10	>10	>10	>10	>10
BPGV05875	MIC	>10	10	>10	5	5	>10	>10	2.5
	MBC	>10	>10	>10	>10	>10	>10	>10	>10
BPGV11111	MIC	>10	10	>10	5	>10	>10	>10	2.5
	MBC	>10	>10	>10	>10	>10	>10	>10	>10
BPGV7301	MIC	>10	10	>10	2.5	10	>10	>10	2.5
	MBC	>10	>10	>10	>10	>10	>10	>10	>10
BPGV07480	MIC	>10	10	>10	2.5	10	>10	>10	2.5
	MBC	>10	>10	>10	>10	>10	>10	>10	>10
BPGV04542	MIC	>10	10	>10	10	10	>10	>10	2.5
	MBC	>10	>10	>10	>10	>10	>10	>10	>10
BPGV05888	MIC	>10	>10	>10	5	>10	>10	>10	2.5
	MBC	>10	>10	>10	>10	>10	>10	>10	>10
BPGV06987 A	MIC	>10	10	>10	5	>10	>10	>10	5
	MBC	>10	>10	>10	>10	>10	>10	>10	>10
BPGV06987 B	MIC	>10	10	>10	5	>10	>10	>10	10
	MBC	>10	>10	>10	>10	>10	>10	>10	>10
BPGV02906	MIC	>10	5	>10	5	>10	>10	>10	2.5
	MBC	>10	>10	>10	>10	>10	>10	>10	>10
Streptomicin 1 mg/mL	MIC	0.007	0.01	0.06	0.007	0.007	0.007	0.007	0.007
	MBC	0.007	0.01	0.06	0.007	0.007	0.007	0.007	0.007
Methicilin 1 mg/mL	MIC	n.d.	n.d.	n.d.	n.d.	n.d.	n.d.	n.d.	0.007
	MBC	n.d.	n.d.	n.d.	n.d.	n.d.	n.d.	n.d.	0.007
Ampicillin 10 mg/mL	MIC	0.15	0.15	0.63	0.15	0.15	n.d.	0.15	0.15
	MBC	0.15	0.15	0.63	0.15	0.15	n.d.	0.15	0.15

MIC: Minimum Inhibitory Concentration; MBC: Minimum Bactericidal Concentration; not detected (n.d.).

## Data Availability

The original contributions presented in the study are included in the article/Appendix A, further inquiries can be directed to the corresponding author.

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
