# Peer review of "Chemical Profile and Biological Activities of Brassica rapa and Brassica napus Ex Situ Collection from Portugal"

_foods, 2024, doi:10.3390/foods13081164_

Round 1

Reviewer 1 Report

Comments and Suggestions for Authors

The present work by Serrano et al., features an analysis the glucosinolates and phenolic content of a Turnip collection ex situ preserved in Portugal. In addition, the antioxidant and antimicrobial activities were evaluated. The technical details, such as the methods used, how they were performed, and description are worthy of commendation. The results section was also properly presented in light of the data obtained. Taken together, the work was a modest attempt at identifying potentially superior accessions with respect to their health benefits for further breeding purposes. That said, the work has quite a number of substantive limitations.

The first is that the aim is not clearly articulated (even in the Abstract). This needs to be improved for clarity.

Similarly, the Introduction is exceedingly lengthy without a clear statement of objective or purpose. What was the expected or anticipated benefit of conducting this research?

This weakness was also reflected in the Conclusions section, where there was no indication of the implication of the findings, especially with respect to the area of Foods. 

The title phrase “Analysis Brassica Biodiversity” is apparently misleading and should either be removed from the title or modified.

Abstract.

-“B. rapa” should be written in full since it is the first time the name is being mentioned. Subsequently, it is acceptable to abbreviate.

-“The extracts exhibited strong activity against S. enterica and S. aureus (MIC values: 5 – 2.5 mg.mL-1), and two of them showed inhibitory activity against E. coli and Y. enterocolitica (MIC values: 5 mg.mL-1).”

The underlined claim is questionable in view of the MIC values reported. What standard guideline did the authors use in arriving at this conclusion?

-The Introduction section is written in a manner that is clear and detailed, providing the reader with a lot of information. Most of the information provided is relevant whereas others are not. In other words, the Introduction section is too lengthy and should be shortened. Only information that is relevant to the topic is expected to be provided.

The main limitation in this work is the failure to clearly articulate the objectives and the hypothesis or assumptions that warranted this research. Also, there is an absence of the relevance and potential implications of this work in the area of food. This should be robustly addressed in every aspect of the manuscript, i.e., Abstract, Introduction, Results, Discussion and Conclusion.

2.5.2. Include the name of the medium in which the bacteria were cultured.

The Conclusions section provided a summary of the main findings. Authors are encouraged to rather explain the implications of these findings, especially in areas related to food.

-The writeup is clear to read and easy to understand. However, major errors in terms of the structure, flow, grammar, spelling, syntax pervade the entire manuscript. As such, authors are strongly encouraged to seek for professional assistance with respect to editing the work.

Comments on the Quality of English Language

The mansucript replete with errors. A robust editing of English language as well as the structure and flow of the writeup by a professional is recommended.

Author Response

Revisor #1

Author’s Notes to Reviewer #1:

All suggestions proposed along the document by the reviewer were accepted and modified.

The first is that the aim is not clearly articulated (even in the Abstract). This needs to be improved for clarity.

Answer: The authors changed the abstract, shortened the introduction and modified the conclusions considering the reviewer's comments and suggestions.

Similarly, the Introduction is exceedingly lengthy without a clear statement of objective or purpose. What was the expected or anticipated benefit of conducting this research?

This weakness was also reflected in the Conclusions section, where there was no indication of the implication of the findings, especially with respect to the area of Foods.

Answer: The objectives of this research, and the potential implications in food area has been incorporated in the manuscript, i.e., Abstract, Introduction, Results, Discussion and Conclusion.

The title phrase “Analysis Brassica Biodiversity” is apparently misleading and should either be removed from the title or modified.

Answer: The title was modified for “Chemical Profile and Biological Activities of Brassica rapa and Brassica napus Collection Ex Situ from Portugal”.

Abstract

-“B. rapa” should be written in full since it is the first time the name is being mentioned. Subsequently, it is acceptable to abbreviate.

Answer: In the Abstract “B. rapa” was written in full.

-“The extracts exhibited strong activity against S. enterica and S. aureus (MIC values: 5 – 2.5 mgmL-1 ), and two of them showed inhibitory activity against E. coli and Y. enterocolitica (MIC values: 5 mg.mL-1 ).”

The underlined claim is questionable in view of the MIC values reported. What standard guideline did the authors use in arriving at this conclusion?

Answer: The classification of MIC values of an extract can be determined based on a comparison with standard reference MIC values or by using criteria that have been proposed by various researchers. Our classifications were compared with Aligiannis et al., 2001, where determine that MIC values for plant extracts based on the literature are:

Strong Inhibitor: Plant extracts that have MIC values less than or equal to 0.5 mg/mL are often considered to be strong inhibitors of microbial growth.

Moderate Inhibitor: MIC values ranging from >0.5 mg/mL to 1.0 mg/mL are typically classified as moderate inhibitors.

Weak Inhibitor: Extracts with MIC values greater than 1.0 mg/mL up to 2.0 mg/mL may be considered weak inhibitors.

Ineffective: If the MIC value is above a certain threshold, commonly around 8.0 mg/mL or greater, the extract may be deemed ineffective or having no significant inhibitory effect.

It's important to note that these categories are not fixed and can vary depending on the specific protocols and criteria established by the researchers or standards. Also, different microorganisms and different extracts may have different thresholds for these classifications.

Aligiannis, N., Kalpoutzakis, E., Mitaku, S., & Chinou, I. B. (2001). Composition and antimicrobial activity of the essential oils of two Origanum species. Journal of agricultural and food chemistry, 49(9), 4168-4170.

- The Introduction section is written in a manner that is clear and detailed, providing the reader with a lot of information. Most of the information provided is relevant whereas others are not. In other words, the Introduction section is too lengthy and should be shortened. Only information that is relevant to the topic is expected to be provided.

Answer: The authors shortened the introduction and tried to provide only the necessary information for the reader.

The main limitation in this work is the failure to clearly articulate the objectives and the hypothesis or assumptions that warranted this research. Also, there is an absence of the relevance and potential implications of this work in the area of food. This should be robustly addressed in every aspect of the manuscript, i.e., Abstract, Introduction, Results, Discussion and Conclusion

Answer: The objectives of the work were specified in every aspect of the manuscript Abstract, Introduction, Results, Discussion and Conclusion.

2.5.2. Include the name of the medium in which the bacteria were cultured.

Answer: These microorganisms were incubated at 37 °C in TSB (Tryptic Soy Broth) fresh medium for 24 h before analysis to ensure maintain they were in the exponential growth phase.

The Conclusions section provided a summary of the main findings. Authors are encouraged to rather explain the implications of these findings, especially in areas related to food.

Answer: The authors include the main findings in food area in the Conclusion section.

-The writeup is clear to read and easy to understand. However, major errors in terms of the structure, flow, grammar, spelling, syntax pervade the entire manuscript. As such, authors are strongly encouraged to seek for professional assistance with respect to editing the work.

Answer: The authors requested help from colleagues whose first language is English.

Reviewer 2 Report

Comments and Suggestions for Authors

The manuscript presents an experimental approach that could benefit from further methodological rigor. It is suggested that the article undergoes a thorough review to address the experimental concerns before considering it.

More specific:

Where are the line numbers?

Abstract in one paragraph.

Bacillus cereus in italics.

Reduce short paragraphs in the introduction. Combine them.

Make the aim of the scope stronger.

Increase the resolution of Figure 1.

Very confused methodology!

S2.3: material-to-liquor or solid-to-liquid?

S2.5.1: essential oil? You didn’t mention this before.

Tables 3 and 4: Why do you have 2 different columns for TPC? What was the difference per 100 g? In DPPH it should be mmol TE/g.

Figure 3: What are the oval shapes?

Conclusions should be rewritten and emphasize the main findings.

Most of the references are old!

Comments on the Quality of English Language

English very difficult to understand/incomprehensible.

Author Response

Revisor #2

Author’s Notes to Reviewer #2:

All suggestions proposed along the document by the reviewer were accepted and modified.

The manuscript presents an experimental approach that could benefit from further methodological rigor. It is suggested that the article undergoes a thorough review to address the experimental concerns before considering it.

Answer: The authors changed the methodology of the considering the reviewer's comments and suggestions.

More specific:

Where are the line numbers?

Answer: The manuscript was submitted with line numbers.

Abstract in one paragraph.

Answer: The Abstract was modified as suggestion.

Bacillus cereus in italics.

Answer: Bacillus cereus was written in italics.

Reduce short paragraphs in the introduction.

Combine them. Make the aim of the scope stronger.

Answer: The Introduction was modified and the paragraphs were reduced and the objectives of the work are stronger.

Increase the resolution of Figure 1.

Answer: It wasn't possible to enhance the image for better resolution, so the decision was made to delete it.

S2.3: material-to-liquor or solid-to-liquid?

Answer: S2.3: The methodology was corrected and the right term “solid-to-liquid” was replaced.

S2.5.1: essential oil? You didn’t mention this before.

Answer: S2.5.1: The methodology was corrected and the term “essential oil” was removed.

Very confused methodology!

Tables 3 and 4: Why do you have 2 different columns for TPC? What was the difference per 100 g? In DPPH it should be mmol TE/g.

Answer: The two columns for TPC refers to:

The first column result refers to the gallic acid equivalents (mg GAE/g) that exist in one gram of dry extract weight.

The second column result refers to the equivalent grams of gallic acid, which exist in 100 grams of the dry plant. The decision was to remove this column.

DPPH units have been corrected in the table.

Figure 3: What are the oval shapes?

Answer: The different colours represent different districts and the circles group the accessions by Portugal region, trying to define particular combinations of geographical profiles/accessions/GLS. GLS. The text does not clearly reference each of the circles. It is considered that keeping the circles in the image does not aid the intended interpretation. The decision was made to replace the image without adding circles

Conclusions should be rewritten and emphasize the main findings.

Answer: The authors include the main findings in food area in the Conclusion section.

Most of the references are old!

Answer: The work done in this area is reported to 2000.

English very difficult to understand/incomprehensible.

Answer: The authors requested help from colleagues whose first language is English.

Reviewer 3 Report

Comments and Suggestions for Authors

This is an interesting article which analyzed Brassica Biodiversity, presenting the  chromatographic and biological perspectives on methanol extracts of a turnip collection ex situ conserved in Portugal. However, there are some ambiguities that need to be corrected/completed:

- the aim of the study is not clearly stated

L39 & 40: "mg.mL-1" remove the dot after the mass unit - it should have been a multiplication sign, but a space or fractional line should be used.

L55: add new data (at least 2022 should be included)

L67. L77;  same references cited in two forms (as numbers and in the form (name, year)?

L80: what means "as shown" Where?

Figure 1 must be enlarged to a minimum of 200% in order to see something

L261: it is r2

L273: it should be "°C"

In table 2 are used some nonstandard features (such as F2b, etc) which should then be explained below the table

Table 3 - (L419), how do you mean -different letters in the same row when the parameter, which can be compared, is presented in the column?

I can't connect the statement below the table with the one in the table: "Access results sorted by date of collection"

Table 4 - again are mentioned letters which indicate significant differences  

Table 5. state below the table what n.t. means.

Figure 3. The title is too general

Figure 4: based on the spaces in distance line (Euclidean distance axis)- the first value should be 0e+08 or 0 

title Fig.4 . Why now r instead of r2

the title and conclusion are not directly related, and part of the reason is that it is not clearly defined what is the primary goal of the work.

Sincerely

Author Response

Revisor #3

Author’s Notes to Reviewer #3:

All suggestions proposed along the document by the reviewer were accepted and modified.

- the aim of the study is not clearly stated

Answer: The authors changed the objective of the study to make it clearer.

L39 & 40: "mg.mL " remove the dot after the mass unit - it should have been a multiplication sign, but a space or fractional line should be used.

Answer: The dot after the mass unit was replaced by a space.

L55: add new data (at least 2022 should be included)

Answer: The authors present a vast bibliography including articles from 2023.

L67. L77; same references cited in two forms (as numbers and in the form (name, year)?

Answer: The reference in the form (name, year) was removed.

L80: what means "as shown" Where?

Answer: It was a mistake, and was deleted

Figure 1 must be enlarged to a minimum of 200% in order to see something.

Answer: It wasn't possible to enhance the image for better resolution, so the decision was made to delete it.

L261: it is r2

Answer: L261: Was corrected.

L273: it should be "°C"

Answer: Was corrected.

In table 2 are used some nonstandard features (such as F2b, etc) which should then be explained below the table.

Answer:  The significance of the symbols F1, F2a and F2b was introduced as footnote of Table 2 Section 3.1 provides a thorough explanation of the meaning of these unusual symbols.

 Table 3 - (L419), how do you mean -different letters in the same row when the parameter, which can be compared, is presented in the column?

Answer: ….”different letters in the same column” was a mistake that was corrected.

I can't connect the statement below the table with the one in the table: "Access results sorted by date of collection"

Answer: The statement below table 4 was removed.

Table 4 - again are mentioned letters which indicate significant differences.

Answer: ”different letters in the same column” was a mistake that was corrected.

Table 5. state below the table what n.t. means.

Answer: Table 5. state below the table must be n.d. (not detectable). It was a mistake.

Figure 3. The title is too general.

Answer: Enhanced subtitle, done.

Figure 4: based on the spaces in distance line (Euclidean distance axis)- the first value should be 0e+08 or 0.

Answer: Figure 4 is included as requested.

title Fig.4 . Why now r instead of r2.

Answer: To validate dendrograms in hierarchical methods, the cophenetic correlation coefficient (r) can be used. This coefficient measures the degree of fidelity with which a dendrogram preserves the pairwise distances between the original unmodelled data points. It is important that the value of the cophenetic correlation coefficient is greater than 0.80.

The title and conclusion are not directly related, and part of the reason is that it is not clearly defined what is the primary goal of the work

Answer: The title was modified to “Chemical Profile and Biological Activities of Brassica rapa and Brassica napus Collection Ex Situ from Portugal” and the Conclusion was related according to the primary goal of the work.

Reviewer 4 Report

Comments and Suggestions for Authors

Analyzing Brassica Biodiversity: Chromatographic and Biological Perspectives on Methanol Extracts of a Turnip Collection Ex Situ conserved in Portugal

The manuscript deals with an interesting subject, namely the characterization of 29 accessions of B. rapa (turnips) and 9 B. napus (turnip and seed) species from different regions of Portugal. The authors analyze parameters such as: glucosinolates (GLS), phenolic compounds, antioxidant and antimicrobial activities with the aim of determining their active principles.

The study is very complex and has interesting results that could be published if the writing of this paper and the the researches are improved. There are some errors, omissions (in my opinion) in the manuscript and in its current form more explanations and/or changes are needed. Overall, the article is well written and presents a topic relevant to academia. Overall experiments are well done, and the results support the conclusions (but which in my opinion must be changed).

I take this opportunity to congratulate the authors for their work.

I have written some suggestions as a way to further improve the study. Specific comments are provided below:

 The title of the article

 Line 2-4: I suggest changing and shortening the title of the article. It's too grandiose. Be shorter and more concise in writing it. E.g: Chemical Profile and Biological Activities of Brassica rapa and B. napus Collection Ex Situ from Portugal

Abstract

The journal's recommendation is that the abstract should be a total of about 200 words maximum. Try to reduce it! Start with the following changes:

Line 32: Remove FRAP and DPPH. Only Antioxidant activity remains in the text.

Line 34-37: Remove ATCC item number. It is sufficient to specify the bacteria in the abstract. They can be found anyway in point 2.5.2.

Line 35: Salmonella enterica subsp - which subspecies? It must be specified

Line 36: Bacillus cereus – in italics

Line 39 and 40: Delete the brackets with the value MIC from the abstract.

Introduction

 Page 2 and 3: The introduction is quite extensive. Relevant information are presented, but it is too long. Try to shorten it!

Line 53: The second paragraph can be presented as a continuation of the first. End the sentence with a punctuation mark: period/ "full stop".

Line 67: The bibliography is cited twice. Delete the bibliographical note cited as (Esteve M, 2020) from the text. There is position [9] for him.

Line 70: The size of the text characters, located in the first row, is different. “Brassica seeds, due to their….”

Line 77-78: The bibliography is cited twice. Delete bibliographic notes cited as (Prakash et al…….) from the text. There is position [8,14] for them.

Line 86: B. rapa in italic.

Line 100: Delete: "in Asia, Europe, and North America" from the text. The rest can remain.

Line 112-113: Delete: “who descended from Western Europe several centuries B.C”. The rest of the text can remain.

Line 129-131: Delete the entire sentence: “The BPGV has conserved a collection of vegetables from the Mediterranean diet, which gained importance when issues of healthy eating and obesity became priorities.”

Line 135 and 156-178: Regarding figure 1, if it shows only the areas where farmers cultivate local varieties of Brassica species, and not the sampling point, it can be deleted. It is sufficient to specify in the text the central-northern areas where it is cultivated. In addition, Table 1 shows the region where the samples come from. Anyway, it's not OK to end a chapter with a figure.

 Materials and Methods

 Line 180: At point 2.1. you only have chemicals, not materials. Plant materials are at point 2.2.

I suggest deleting subchapter 2.1. The chemical reagents appear in the analysis methods anyway and you can specify their producer in the text. Anyway, the expression in the text...where the reagents were procured, is particularly unsatisfactory.

Line 200: I suggest that after Braga, Minho province should be specified - this for an easier localization.

Line 239: -1 to the superscript.

Line 261: to the correlation coefficient, 2 to superscript

Line 266: Delete FRAP from title or text.

Line 272: EO is the abbreviation of essential oil? It is not specified.

Line 273: space between 4 and min

Line 239, 273: Check if the Degree Symbols “ ° “ as in text format.

Line 281: I don't think it is correct to include TPC in point 2.5.1. Antioxidant activity. It is only a (important) chemical parameter responsible for the antioxidant activity.

Line 298, 436, 450, 455, 459, 464, 471, 481, 484, 491, 495, 529, table 5 (590): Enterobacter cloacae; Line 432, 452, 457, 460, 466, 468, 473, 479, 486, 493, 497, 501, 517, 534, table 5 (590), 721: Escherichia coli, …. etc. The names of the microorganisms must appear without abbreviation, when they appear for the first time in the text. Afterwards, it must appear only with abbreviations, which is not respected.

Line 303: which fresh medium was used?

Line 315: “Ampicillin and Streptomicin were used” - were both antibiotics used in the same sample? And in what concentrations?

Line 316: what was the concentration for Methicillin?

Line 321: which solid medium was used?

 Results and discussions

 Line 339: At point 3. Results add and discussions in the chapter title.

Line 348: Figure 1 ? Check if it is correct?

Line 353: [R+ C2NHO3S ] and [R+ CS2O3]  - Put the valences in the subscript

Line 363-364: Put the phrase as a footnote to the table: The symbols F1 and F2a/b indicate fragmentation paths which are specific to each GLS molecular structure (differentiated by the R functional group”

Line 365…: Reorganize the 2nd table, align the content of the first columns vertically, and the content of the last one to remain horizontally

Line 366: Delete the bracket (FRAP and DPPH•), respectively (TPC) or Total Phenolica Content. It is specified in point 2.5.1. the fact that DPPH and FRAP assays are used to estimate antioxidant activities contents, and in the case of Total Phenolic Content the abbreviation is suitable only after the first use in the text.

Line 380: At the beginning of the sentence: TPC instead of Total Phenolic Content.

Use “antioxidant capacity” or “antioxidant activity” or “antioxidant power” instead of “antioxidant activity power”. This is not a common term.

Line 393: When you specify a negative correlation, do you mean a correlation coefficient between 0 and -1 (less than zero signifies a negative relationship) or a low positive correlation coefficient?

Line 414 and 422: At Table 3 and 4, delete the bracket (FRAP and DPPH•), respectively (TPC) or total phenolic content.

Line 414 and 422: Please specify why in Table 3 and 4 there are two columns for TPC, per g and per 100g? In the text I see you interpret the values only using GAE/g. Explain or modify!

R430-531: The informations from Table 5 are repeated and there are very long. It must be reduced and only what is essential should remain

Line 522: E. coli and Salmonella enterica in italic

Line 590: Table 5 does not follow the format indicated by the journal.

Line 590: Add as a footnote the meaning of n.t. in table 5.

Line 597: B. in italics and not underlined.

Line 700: I suggest you insert Figure 4 in the text. A chapter should not end with a figure, but with its description.

 Conclusions

 Line 708-713 and 718: The conclusions seem discussions. In conclusions, it is not good to present the bibliography or refer to tables. Only the conclusions of the research should be specified, without sending the reader to the references.

 References

 Line 748: The bibliography is written carelessly. Almost all references must be corrected. Please follow the journal requirements. First of all, there is no Abbreviated Journal Name. At the same time, between positions 18 and 19, there is a bibliographic note no. 1, etc.

 I have no other objections.

Please, try to improve it!

Comments on the Quality of English Language

There are little infelicities of speech, which is why the article must be seen by a native English speaker.

Author Response

Revisor #4

Author’s Notes to Reviewer #4:

All suggestions proposed along the document by the reviewer were accepted and modified.

Abstract

Answer: Line 2-4: The title was modified according suggestion of the reviewer.

Answer: The Abstract was reduced to a total of about 206 words maximum.

Answer: Line 32: The words FRAP and DPPH were removed.

Answer: Line 34-37: The ATTC item number was removed from the Abstract.

Answer: Line 35: Salmonella enterica subsp. enterica serovar enteritidis ATCC 13076.

Answer: Line 36: Bacillus cereus was has been italicized.

Answer: Line 39 and 40: The brackets with the value MIC were removed from the abstract

Introduction

Answer: Page 2 and 3: The Introduction was shortened and only relevant information is presented.

Answer: Line 53: The sentence was corrected.

Answer: Line 67: The bibliography cited twice was removed.

Answer: Line 70: The size of the text characters, located in the first row, is different. “Brassica seeds, due to their….” We don’t understand what the reviewer wants….

Answer: Line 77-78: The bibliography cited twice was deleted.

Answer: Line 86: B. rapa in italic. The word was has been italicized.

Answer:  Line 100: The text "in Asia, Europe, and North America" was deleted.

Answer: Line 112-113: The text “who descended from Western Europe several centuries B.C”, was deleted.

Answer: Line 129-131: The text “The BPGV has conserved a collection of vegetables from the Mediterranean diet, which gained importance when issues of healthy eating and obesity became priorities”, was deleted.

Answer: Line 135 and 156-178: It wasn't possible to enhance the image for better resolution, so the decision was made to delete it

Answer: Line 180: Chemicals have been removed from subchapter 2.1

Answer: The subchapter 2-1 was deleted and the changes were made as suggested by the reviewer.

Answer: Line 200: The suggestion was done” in Braga, North-west of Minho.

Answer: Line 239: It was changed as suggested.

Answer: Line 261:It was changed as suggested.

Answer: Line 266: It was changed as suggested.

Answer: Line 272: EO was removed from the text.

Answer: Line 273: It was changed as suggested.

Answer: Line 239, 273: It was changed as suggested.

Answer: Line 281: We agree, however some authors consider TPC as an indirect measure of antioxidant activity, and justifies the antioxidant activity being due only to GLS compounds.

Answer: Line 298, 436, 450, 455, 459, 464, 471, 481, 484, 491, 495, 529, table 5 (590): Enterobacter cloacae; Line 432, 452, 457, 460, 466, 468, 473, 479, 486, 493, 497, 501, 517, 534, table 5 (590), 721: Escherichia coli, …. etc. It was changed as suggested.

Answer: Line 303: The fresh medium is TSB (Tryptic Soy Broth).

Answer: Line 315: Methicillin was uses only for Staphylococcus aureus. Ampicillin was used for all bacteria except for Bacillius cereus. The concentrations are indicated in the text and in table 5.

Answer: Line 316: Methicillin was tested at 1mg/mL

Answer: Line 321: The solid medium used was blood agar

Results and discussions

Answer: Line 339: At point 3. The text was changed according to the proposal.

Answer: Line 348: In fact, Figure 1 is not related to the GLS profiles. Due to an undetected error, Figure 2 showing the general structure of a GLS molecule was not introduced in the submitted document. The figure, has now been included in the revised version of the manuscript, denoted as Figure 1

Answer: Line 353: The formulas were changed according to the proposal.

Answer: Line 363-364 The phrase was introduced as a footnote of Table 2.

Answer: Line 365: The Table 2 was again formatted according to your suggestion

Answer: Line 366: The text was changed according to the proposal.

Answer: Line 380: The text was changed according to the proposal.

Answer: Use “antioxidant capacity” or “antioxidant activity” or “antioxidant power” instead of “antioxidant activity power”. This is not a common term. The text was changed according to the proposal.

Answer: Line 393: We want to specify a low positive correlation coefficient.

Answer: Line 414 and 422: The text was changed according to the proposal.

Answer: Line 414 and 422:

The first column result refers to the gallic acid equivalents (mg GAE/g) that exist in one gram of dry extract weight.

The second column result refers to the equivalent grams of gallic acid, which exist in 100 grams of the dry plant. The decision was to remove this column.

Answer: R430-531: Regarding the size of the table, we cannot summarize it, as there are many samples.

Answer: Line 522: The words were italicized.

Answer: Line 590: Table 5 was changed according the format indicated by the journal.

Answer: Line 590: The footnote in table 5 means not detectable (n.d.), and was changed according the proposal.

Answer: Line 597: B. in italics and not underlined.

Answer: Line 700: Figure 4 was insert in the text.

Conclusions

Answer: Line 708-713 and 718: The conclusions were modified according the reviewer suggestions.

References

Answer: Line 748: The bibliography was corrected.

Answer: The authors requested help from colleagues whose first language is English.

Round 2

Reviewer 1 Report

Comments and Suggestions for Authors

Authors have addressed all the questions raised. The manuscript is acceptable in its current form.

Comments on the Quality of English Language

English language is mostly fine. However, there were so many instances where sentences were made into paragraphs when there was no need to.

Author Response

Revisor #1- Round 2

Author’s Notes to Reviewer #1:

We would like to extend our sincere appreciation for taking the time to review our manuscript and for providing insightful suggestions for improvement. Your feedback has been immensely valuable in refining our work.

We want to express our gratitude for your thoughtful recommendations, which we have diligently considered and endeavored to incorporate into the document. Your expertise and constructive criticism have undoubtedly strengthened the quality of our research.

Thanks,

Reviewer 2 Report

Comments and Suggestions for Authors

The paper has been revised according to the suggestions and criticisms of the reviewers. In this revised version, the paper has improved its quality and I recommend the article for publication.

Author Response

Revisor #2- Round 2

Author’s Notes to Reviewer #2:

We would like to extend our sincere appreciation for taking the time to review our manuscript and for providing insightful suggestions for improvement. Your feedback has been immensely valuable in refining our work.

We want to express our gratitude for your thoughtful recommendations, which we have diligently considered and endeavored to incorporate into the document. Your expertise and constructive criticism have undoubtedly strengthened the quality of our research.

Thanks,

Reviewer 3 Report

Comments and Suggestions for Authors

Although the authors did not attach a new version of the work with marked parts that were changed, by comparing the documents (v1 vs v2) I found that the requested corrections were implemented. With this, the article gained in quality and now I can recommend it for publication.

Author Response

Revisor #3- Round 2

Author’s Notes to Reviewer #3:

Thank you very much for your thorough review and for taking the time to compare the initial and revised versions of our manuscript. We are delighted to hear that the requested corrections have been successfully implemented and that you now find the article to be of higher quality.

We apologize for not providing a marked-up version of the changes to the manuscript, but it would be too confusing. Your feedback has been duly noted and we will ensure we include such documentation in any future reviews to facilitate the review process.

Your recommendation for publication is truly appreciated and serves as a validation of our efforts to address the issues raised during the review process.

Once again, we sincerely thank you for your valuable input and support throughout this endeavor.

Thanks,

Reviewer 4 Report

Comments and Suggestions for Authors

The author has made revisions to address all the issues pointed out in the first review. The revised paper has a rigorous logic and an accurate narration. The article is well written. Readability has benefited enormously from the changes made, especially the additional clarifications.

I no longer have any clarification related to the content of the article, but only related to the figure no. 1. I don’t think it is appropriate to present figure no. 1. I would eliminate it! It seems too pedagogical / didactic to me... it's too trivial.

Congratulations to the authors for the improved revised manuscript!

Comments on the Quality of English Language

The paper must be seen by a native English speaker.

Author Response

Revisor #4 - Round 2

Author’s Notes to Reviewer #4:

Thank you very much for your thorough evaluation of our revised manuscript and for acknowledging the improvements made. We greatly appreciate your positive feedback regarding the rigorous logic, accurate narration, and enhanced readability of the article.

We are pleased to hear that the revisions have addressed the issues pointed out in the initial review and have contributed to the overall quality of the manuscript.

Regarding your comment on Figure 1, we appreciate your perspective and your suggestion to eliminate it. However, given that the manuscript was submitted to a special issue that covers several areas of food research, we believe it is important to maintain Figure 1. The structure of a glucosinolate may not be as straightforward for researchers in fields that are not directly related to Chemistry. Moreover, the discussion regarding the fragmentation process that leads to the fragments F1 and F2a/F2b in section 3.1 becomes more evident through the GLS structure presents in the Figure 1.

Thank you for highlighting the importance of a thorough language review. We will take your advice into account and seek assistance from a native English speaker to further refine the manuscript's language.

Once again, we express our gratitude for your valuable feedback and for recognizing the efforts we have put into improving the manuscript.

Thanks
